# ENCODING RECURRENCE INTO TRANSFORMERS

**Feiqing Huang**[1][*][†]**, Kexin Lu**[1][*]**, Yuxi Cai**[1]**, Zhen Qin**[2]**, Yanwen Fang**[1]
**Guangjian Tian**[2]**, Guodong Li**[1][‡]
Department of Statistics and Actuarial Science, The University of Hong Kong[1]
Huawei Noah's Ark Lab[2]
`{amieehuang,neithen,caiyuxi,u3545683}@connect.hku.hk`
`gdli@hku.hk, {qin.zhen, tian.guangjian}@huawei.com`

## ABSTRACT

This paper novelly breaks down with ignorable loss an RNN layer into a sequence of simple RNNs, each of which can be further rewritten into a lightweight positional encoding matrix of a self-attention, named the Recurrence Encoding Matrix (REM). Thus, recurrent dynamics introduced by the RNN layer can be encapsulated into the positional encodings of a multihead self-attention, and this makes it possible to seamlessly incorporate these recurrent dynamics into a Transformer, leading to a new module, Self-Attention with Recurrence (RSA). The proposed module can leverage the recurrent inductive bias of REMs to achieve a better sample efficiency than its corresponding baseline Transformer, while the self-attention is used to model the remaining non-recurrent signals. The relative proportions of these two components are controlled by a data-driven gated mechanism, and the effectiveness of RSA modules are demonstrated by four sequential learning tasks.

## 1 INTRODUCTION

Sequential data modeling is an important topic in machine learning, and the recurrent networks such as LSTM (Hochreiter & Schmidhuber, 1997) and GRU (Chung et al., 2014) have served as the benchmarks in this area over a long period of time. The success mainly contributes to the variety of recurrent dynamics introduced by these models, referred to as the recurrent inductive bias. More specifically, the dependence between any two inputs can be described by a parametric form, which heavily depends on their relative temporal locations. However, the recurrent models are well known to suffer from two drawbacks. The first one is the gradient vanishing problem (Hochreiter et al., 2001), i.e. the recurrent models have difficulty in depicting the possibly high correlation between distant inputs. This problem cannot be solved fundamentally by the recurrent models themselves, although it can be alleviated to some extent, say by introducing long memory patterns (Zhao et al., 2020). Secondly, the sequential nature renders these models difficult to be trained in parallel (Vaswani et al., 2017). In practice, many techniques were proposed to improve the computational efficiency of recurrent models, while they all come with compromises (Luo et al., 2020; Lei et al., 2017).

In recent years, Transformers (Vaswani et al., 2017) have been revolutionizing the field of natural language processing by achieving the state-of-the-art performance on a wide range of tasks, such as language modeling (Kenton & Toutanova, 2019), machine translation (Dai et al., 2019) and text summarization (Liu & Lapata, 2019), etc. They have also demonstrated great potential in other types of sequence learning problems, for instance, time series forecasting (Zhou et al., 2021; Li et al., 2019). The success of Transformers is due to the fact that the similarity between any two tokens is well taken into account (Vaswani et al., 2017) , and hence they can model long range dependence effortlessly. Moreover, contrary to the recurrent models, the self-attention mechanism in Transformers is feed-forward in nature, and thus can be computed in parallel on the GPU infrastructure (Vaswani et al., 2017). However, the flexibility also leads to sample inefficiency in training a Transformer, i.e. much more samples will be needed to guarantee good generalization ability (d'Ascoli et al., 2021). Moreover, the chronological orders are usually ignored by Transformers since they are time-invariant, and some additional efforts, in the form of positional encoding, will be required to further aggregate the temporal information (Shaw et al., 2018; Vaswani et al., 2017; Dai et al., 2019).

---

[*]Equal contribution. [†] This work was done during the author's internship at Huawei Noah's Ark Lab.
[‡]Correspondence to gdli@hku.hk.

Figure 1: (a) – (c) plot the two data features, namely the sample signal and the sample size as the $x$ and $y$ axis, respectively. The model performance in each data region for the RNN, Transformer and RSA are given, where deeper color implies a better performance. (d) In each attention head, the proposed RSA attaches an REM to a normalized self-attention score via a gated mechanism, with gate value $\sigma(\mu)$. The REM depicts a type of recurrence dependence structure between the tokens in $\boldsymbol{X}$, and is parameterized by one or two parameters, i.e. $\lambda$ or $(\gamma, \theta)$, where $\lambda = \tanh(\eta), \gamma = \sigma(\nu)$.

In short, both recurrent and Transformer models have the pros and cons in modeling sequential data. On one hand, due to inductive bias, the recurrent models excel at capturing the recurrent patterns even with relatively small sample sizes; see Figure 1(a). Meanwhile, sample size is the performance bottleneck for the Transformer models and, when there are sufficient samples, they are supposed to be able to depict any recurrent or non-recurrent patterns in the data; see Figure 1(b). On the other hand, sequential data have recurrent patterns more or less, and Transformers may have an improved performance if the recurrent model can be involved to handle these patterns, especially when the sample size is relatively small. Specifically, if the recurrent and non-recurrent components are separable, then one can apply a parsimonious recurrent model on the recurrent component and a Transformer on the non-recurrent one. As a result, the sample efficiency can be improved comparing to the Transformer-only baseline; see illustration in Figure 1(c).

There have been various attempts in the literature to combine the two models. Some earlier works were to simply stack them together in a straightforward manner. Chen et al. (2018) mixed and matched a Transformer's encoder with an recurrent-based decoder. Hao et al. (2019) introduced an additional recurrent encoder to a Transformer, while Wang et al. (2019) stacked a recurrent layer prior to the multihead self-attention. These proposals inherit both the aforementioned shortcomings of Transformer and recurrent models. In particular, for a very long input sequence, the sequential operation in the recurrent layers become extremely expensive.

Recent efforts have been spent on integrating recurrence and self-attention systematically. Feedback Transformer (Fan et al., 2021) introduces the memory vectors to aggregate information across layers, and uses them to update the next token in a recursive manner. However, the computationally expensive sequential operation limits its attractiveness. Another line of research applies the recurrent operation only to aggregate the temporal information at a coarser scale, while the token-by-token dependence is learned by self-attention instead. Transformer-XL (Dai et al., 2019) partitions the long inputs into segments and introduces a segment-level recurrence. Meanwhile, Temporal Latent Bottleneck (TLB) (Didolkar et al., 2022) and Block-Recurrent Transformer (BRT) (Hutchins et al., 2022) further divide the segments into smaller chunks, and each chunk is summarized into a few state vectors. A recurrent relation is then formed on the sequence of state vectors. These hierarchical designs are useful to reduce the computational burden, while they overlook recurrent dynamics at a finer scale.

In an attempt to simplify the numerical calculation of RNNs, we found surprisingly that an RNN layer with linear activation can be broken down into a series of simple RNNs with scalar hidden coefficients. Each simple RNN induces a distinct recurrent pattern, and their combination forms the recurrent dynamics of the RNN layer. Hence the calculation time can be greatly reduced by training these simple RNNs in parallel. On top of that, it can be equivalently rewritten into the positional encodings of a multihead self-attention (MHSA). This spontaneously inspires a solution, the multihead Self-Attention with Recurrence (RSA), to combine self-attention with RNN into one single operation while maintaining parallel computation. This solution enables our design to preserve the merits from both Transformer and recurrent models, while their respective shortcomings are avoided. More importantly, it can be used to replace the self-attention of existing networks, such as Transformer XL, TLB and BRT, to further explore recurrent dynamics at the finer scale. Our paper makes three main contributions below.

1. With ignorable approximation loss, we demonstrate that an RNN layer with linear activation is equivalent to a multihead self-attention (MHSA); see Figure 2. Specifically, each attention

head can be used to recover a type of recurrent dependence structure, and multiple heads working in sync replicate the complex temporal dynamics of the RNN layer.

2. In the above MHSA, recurrent dynamics of the RNN layer are encapsulated entirely into the positional encodings, which we name the Recurrence Encoding Matrices (REMs). This makes it possible to add the REMs to any existing Transformer architecture that has the self-attention mechanism, leading to the Self-Attention with Recurrence (RSA) module; see Figure 1(d). A gated mechanism is used to control the proportion of REMs, and the gate value may also be interpreted as the proportion of the recurrent signals in the data.

3. Our experiments on four sequential tasks demonstrate that the proposed RSA module can effectively enhance the baseline Transformers' forecasting power. Moreover, from the fitted gate values, we have an interesting finding that time series data have stronger recurrent signals than those of the regular languages, while the code or natural languages have much weaker recurrent signals.

### 1.1 OTHER RELATED WORKS

**Relative positional encoding** The proposed REMs can provide complementary positional encoding in the RSA module. Essentially, its recurrent pattern conveys relative location information. This property is inherited from the distance-awareness of the RNN. Compared to the relative positional encoding (RPE) in the existing literature, it is more parsimonious and interpretable. The fully learnable RPE proposed by Shaw et al. (2018) requires $O(T^2d)$ parameters, where $T$ is the number of tokens and $d$ is the token dimension. Later, different variants are introduced by Huang et al. (2019) and Dai et al. (2019), which reduce the number of parameters to $O(Td)$ and $O(d)$, respectively. In contrast, our REMs have parametric forms of merely one or two parameters.

**Time series forecasting** Recent literature sees a surge in deep learning models designed specially for time series forecasting tasks, which are not Transformer-based. For instance, DLinear (Zeng et al., 2022) and Autoformer (Wu et al., 2021) decompose the series into trend and non-trend components. Autoformer and SCINet (Liu et al., 2021) further exploit the temporal dependence structure of time series via the autocorrelation function and multi-level convolution operations, respectively. FiLM (Zhou et al., 2022a) and FEDformer (Zhou et al., 2022b) transform the series to the frequency domain to capture meaningful signals while removing noises. This paper, however, differs from these literature in both target and method. We aim to add the recurrent inductive bias to general-purpose Transformers, and hence its performance rely heavily on the baseline Transformer that it modifies.

## 2 RELATIONSHIP BETWEEN RNN AND MULTIHEAD SELF-ATTENTION

This section demonstrates that, without loss of much generality, an RNN layer can be approximated by a series of simple RNNs with scalar (hidden) coefficients, which can be further represented in the form of a multihead self-attention.

### 2.1 BREAKING DOWN AN RNN LAYER

Consider an RNN layer with the input variables $\{\boldsymbol{x}_t \in \mathbb{R}^{d_{\text{in}}}, 1 \leq t \leq T\}$, and it has the form of $\boldsymbol{h}_t = g(\boldsymbol{W}_h\boldsymbol{h}_{t-1} + \boldsymbol{W}_x\boldsymbol{x}_t + \boldsymbol{b})$, where $g(\cdot)$ is the activation function, $\boldsymbol{h}_t \in \mathbb{R}^d$ is the output or hidden variable with $\boldsymbol{h}_0 = 0$, $\boldsymbol{b} \in \mathbb{R}^d$ is the bias term, $\boldsymbol{W}_h \in \mathbb{R}^{d \times d}$ and $\boldsymbol{W}_x \in \mathbb{R}^{d \times d_{\text{in}}}$ are weights. When the activation function is linear, i.e. $g(x) = x$, the RNN becomes

$$\boldsymbol{h}_t = \boldsymbol{W}_h\boldsymbol{h}_{t-1} + \boldsymbol{W}_x\boldsymbol{x}_t, \quad \text{or equivalently} \quad \boldsymbol{h}_t = \sum_{j=0}^{t-1} \boldsymbol{W}_h^j \boldsymbol{W}_x \boldsymbol{x}_{t-j}, \quad (1)$$

Figure 2: Illustration on how an RNN layer can be equivalently represented by a set of simple RNNs and further by a multihead self-attention.

where the bias term $\boldsymbol{b}$ is suppressed for simplicity. Although it has a feedforward form, the RNN cannot be trained in parallel, and this is mainly caused by the power $j$ of recurrent weights $\boldsymbol{W}_h$, where $1 \leq j \leq t-1$. This section makes an effort to block diagonalize $\boldsymbol{W}_h$ such that the RNN at (1) can be broken down into a sequence of simple RNNs with scalar (hidden) coefficients.

**Lemma 1** (Theorem 1 in Hartfiel (1995)). *Real matrices with $R$ distinct nonzero eigenvalues are dense in the set of all $d \times d$ real matrices with rank at most $R$, where $0 < R \leq d$.*

Suppose that the weight matrix $\boldsymbol{W}_h$ has rank $R \leq d$. By Lemma 1, without loss of much generality, we can assume that the nonzero eigenvalues of $\boldsymbol{W}_h$ are all distinct. Specifically, $\boldsymbol{W}_h$ has $r$ real nonzero eigenvalues $\lambda_1, \ldots, \lambda_r$, and $s$ pairs of complex nonzero eigenvalues $\lambda_{r+1}, \ldots, \lambda_{r+2s}$, where $(\lambda_{r+2k-1}, \lambda_{r+2k}) = (\gamma_k e^{i\theta_k}, \gamma_k e^{-i\theta_k})$ for $1 \leq k \leq s$, $i$ is the imaginary unit, and $R = r + 2s$. As a result, we have the real Jordan form, $\boldsymbol{W}_h = \boldsymbol{B}\boldsymbol{J}\boldsymbol{B}^{-1}$, where $\boldsymbol{B} \in \mathbb{R}^{d \times d}$ is invertible and $\boldsymbol{J} \in \mathbb{R}^{d \times d}$ is a block diagonal matrix. It holds that $\boldsymbol{W}_h^j = \boldsymbol{B}\boldsymbol{J}^j\boldsymbol{B}^{-1}$ for all $j \geq 1$, and we can then break down the recurrence induced by $\boldsymbol{W}_h$ into that of the $p \times p$ block matrices in $\boldsymbol{J}$ with $p = 1$ or $2$.

Similar to (1), we define three types of RNNs with linear activation below,

$$\boldsymbol{h}_t^{\mathrm{R}}(\lambda) = \sum_{j=1}^{t-1} \lambda^j \boldsymbol{W}_x^{\mathrm{R}} \boldsymbol{x}_{t-j}, \qquad \boldsymbol{h}_t^{\mathrm{C1}}(\gamma, \theta) = \sum_{j=1}^{t-1} \gamma^j \cos(j\theta) \boldsymbol{W}_x^{\mathrm{C1}} \boldsymbol{x}_{t-j}, \qquad (2)$$

and $\boldsymbol{h}_t^{\mathrm{C2}}(\gamma, \theta) = \sum_{j=1}^{t-1} \gamma^j \sin(j\theta) \boldsymbol{W}_x^{\mathrm{C2}} \boldsymbol{x}_{t-j}$, where the first one corresponds to the real eigenvalues, i.e. the $1 \times 1$ block matrices in $\boldsymbol{J}$, while the last two correspond to the complex eigenvalues, i.e. the $2 \times 2$ block matrices in $\boldsymbol{J}$. Note that each of the three RNNs has the recurrent weights of $\lambda$ or $(\gamma, \theta)$, and its form with a nonlinear activation function is given in the Appendix.

**Proposition 1.** *Suppose that $\boldsymbol{W}_h$ has rank $R = r + 2s \leq d$, and its eigenvalues are defined above. Let $\boldsymbol{h}_{0,t} = \boldsymbol{W}_x \boldsymbol{x}_t$, and then the RNN with linear activation at (1) can be equivalently rewritten into*

$$\boldsymbol{h}_t = \sum_{k=1}^{r} \boldsymbol{h}_t^{\mathrm{R}}(\lambda_k) + \sum_{k=1}^{s} \boldsymbol{h}_t^{\mathrm{C1}}(\gamma_k, \theta_k) + \sum_{k=1}^{s} \boldsymbol{h}_t^{\mathrm{C2}}(\gamma_k, \theta_k) + \boldsymbol{h}_{0,t}.$$

## 2.2 An equivalent MHSA representation

Consider the RNN of $\{\boldsymbol{h}_t^{\mathrm{R}}\}$, and let $\boldsymbol{X} = (\boldsymbol{x}_1, \ldots, \boldsymbol{x}_T)' \in \mathbb{R}^{T \times d_{\mathrm{in}}}$ be an input matrix consisting of $T$ tokens with dimension $d_{\mathrm{in}}$, where the transpose of a matrix $\boldsymbol{A}$ is denoted by $\boldsymbol{A}'$ throughout this paper. We first give the value matrix $\boldsymbol{V}$ by projecting $\boldsymbol{X}$ with a linear transformation, i.e. $\boldsymbol{V} = \boldsymbol{X}\boldsymbol{W}_V$ with $\boldsymbol{W}_V = \boldsymbol{W}_x^{\mathrm{R}\prime} \in \mathbb{R}^{d_{\mathrm{in}} \times d}$, and the relative positional encoding matrix is set to

$$\boldsymbol{P}_{\mathrm{mask}}^{\mathrm{R}}(\lambda) = \begin{pmatrix} 0 & 0 & 0 & \cdots & 0 \\ f_1(\lambda) & 0 & 0 & \cdots & 0 \\ f_2(\lambda) & f_1(\lambda) & 0 & \cdots & 0 \\ \vdots & \vdots & \vdots & \ddots & \vdots \\ f_{T-1}(\lambda) & f_{T-2}(\lambda) & f_{T-3}(\lambda) & \cdots & 0 \end{pmatrix}, \qquad (3)$$

where $f_t(\lambda) = \lambda^t$ for $1 \leq t \leq T-1$. As a result, the first RNN at (2) can be represented into a self-attention (SA) form,

$$(\boldsymbol{h}_1^{\mathrm{R}}, \ldots, \boldsymbol{h}_T^{\mathrm{R}})' = \mathrm{SA}^{\mathrm{R}}(\boldsymbol{X}) = [\mathrm{softmax}(\boldsymbol{Q}\boldsymbol{K}') + \boldsymbol{P}_{\mathrm{mask}}^{\mathrm{R}}(\lambda)]\boldsymbol{V}.$$

We call $\boldsymbol{P}_{\mathrm{mask}}^{\mathrm{R}}$ the recurrence encoding matrix (REM) as it summarizes all the recurrence in $\{\boldsymbol{h}_t^{\mathrm{R}}\}$. For the RNN of $\{\boldsymbol{h}_t^{\mathrm{Ci}}\}$, $i = 1$ or $2$, the REM is denoted by $\boldsymbol{P}_{\mathrm{mask}}^{\mathrm{Ci}}(\gamma, \theta)$, which has the form of (3) with $f_t(\lambda)$ being replaced by $f_t(\gamma, \theta) = \gamma^t \cos(t\theta), i = 1$ or $f_t(\gamma, \theta) = \gamma^t \sin(t\theta), i = 2$ for $1 \leq t \leq T-1$. And the value matrix has the form of $\boldsymbol{V} = \boldsymbol{X}\boldsymbol{W}_V$ with $\boldsymbol{W}_V = \boldsymbol{W}_x^{\mathrm{Ci}\prime} \in \mathbb{R}^{d_{\mathrm{in}} \times d}$. Thus, these two RNNs at (2) can also be represented into SA forms,

$$(\boldsymbol{h}_1^{\mathrm{Ci}}, \ldots, \boldsymbol{h}_T^{\mathrm{Ci}})' = \mathrm{SA}^{\mathrm{Ci}}(\boldsymbol{X}) = [\mathrm{softmax}(\boldsymbol{Q}\boldsymbol{K}') + \boldsymbol{P}_{\mathrm{mask}}^{\mathrm{Ci}}(\gamma, \theta)]\boldsymbol{V} \quad \text{with} \quad i = 1 \text{ or } 2.$$

The remaining term in Proposition 1, $\boldsymbol{h}_{0,t}$ depends on $\boldsymbol{x}_t$ only, and there is no inter-dependence involved. Mathematically, we can represent it into a SA with the identity relative positional encoding matrix. Finally, all the query and key matrices $\boldsymbol{Q}$ and $\boldsymbol{K}$ are set to zero in the above reformulations.

**Proposition 2.** *If the conditions of Proposition 1 hold, then the RNN with linear activation at (1) can be represented into a multihead self-attention (MHSA) with $r + 2s + 1$ heads, where the query and key matrices are zero, and relative positional encoding matrices are $\{\boldsymbol{P}_{\mathrm{mask}}^{\mathrm{R}}(\lambda_k), 1 \leq k \leq r\}$, $\{\boldsymbol{P}_{\mathrm{mask}}^{\mathrm{C1}}(\gamma_k, \theta_k), \boldsymbol{P}_{\mathrm{mask}}^{\mathrm{C2}}(\gamma_k, \theta_k), 1 \leq k \leq s\}$ and an identity matrix, respectively.*

The three simple RNNs at (2) provide different temporal decay patterns: $\boldsymbol{h}_t^{\mathrm{R}}$ provides the regular exponential decay induced by the real eigenvalues $\lambda_k$'s, and $\{\boldsymbol{h}_t^{\mathrm{C1}}, \boldsymbol{h}_t^{\mathrm{C2}}\}$ provide the cyclical damped cosine or sine decay induced by the pair of complex eigenvalues $(\gamma_k e^{i\theta_k}, \gamma_k e^{-i\theta_k})$'s. These temporal decay patterns are further summarized into the regular REM, $\boldsymbol{P}_{\mathrm{mask}}^{\mathrm{R}}(\lambda)$, and cyclical REMs, $\boldsymbol{P}_{\mathrm{mask}}^{\mathrm{C1}}(\gamma, \theta)$ and $\boldsymbol{P}_{\mathrm{mask}}^{\mathrm{C2}}(\gamma, \theta)$, respectively; see Figure 2 for the illustration. From Proposition 2, the combination of these three types of patterns forms the recurrent dynamics of the RNN layer at (1).

For each head, the REM has one or two parameters, and $\boldsymbol{W}_V$ can be regarded as one $d \times d$ learnable matrix. This leads to a parameter complexity of $O(Rd^2)$, and it is slightly larger than that of the RNN at (1) since $R$ is usually much smaller than $d$ (Prabhavalkar et al., 2016). Moreover, the MHSA representation in Proposition 2 gives us a chance to make use of parallel matrix calculation on the GPU hardware; see Appendix D.3 for an illustration of the computational efficiency of REMs.

## 3 ENCODING RECURRENCE INTO SELF-ATTENTION

While the query and key matrices are set to zero in the MHSA representation at Proposition 2, they play a central role in a standard Transformer. This motivates us to propose the Self-Attention with Recurrence (RSA) module to seamlessly combine the strengths of RNNs and Transformers:

$$\mathrm{RSA}(\boldsymbol{X}) = \{[1 - \sigma(\mu)]\mathrm{softmax}(\boldsymbol{Q}\boldsymbol{K}') + \sigma(\mu)\boldsymbol{P}\}\boldsymbol{V} \qquad (4)$$

for each head, where $\boldsymbol{P}$ is a regular or cyclical REM, and $\sigma(\mu) \in [0, 1]$ is a gate with $\sigma$ being the sigmoid function and $\mu$ being the learnable gate-control parameter. Figure 1(d) provides a graphical illustration of one RSA head.

Note that the REMs in Section 2 are all lower triangular matrices, which correspond to unidirectional RNNs. In the meanwhile, for non-causal sequential learning tasks (Graves et al., 2005; Zhou et al., 2016), bidirectional RNNs are usually applied, and accordingly we can define the unmasked versions of REMs. Specifically, the regular REM is $\boldsymbol{P}_{\mathrm{unmask}}^{\mathrm{R}}(\lambda) = \boldsymbol{P}_{\mathrm{mask}}^{\mathrm{R}}(\lambda) + [\boldsymbol{P}_{\mathrm{mask}}^{\mathrm{R}}(\lambda)]'$, and the cyclical REMs are $\boldsymbol{P}_{\mathrm{unmask}}^{\mathrm{Ci}}(\gamma, \theta) = \boldsymbol{P}_{\mathrm{mask}}^{\mathrm{Ci}}(\gamma, \theta) + [\boldsymbol{P}_{\mathrm{mask}}^{\mathrm{Ci}}(\gamma, \theta)]'$ with $i = 1$ and 2. In practice, these REMs will be explosive when $|\lambda| > 1$ or $|\gamma| > 1$. To avoid this problem, we further bound these two parameters by transformations in this paper, i.e. $\lambda = \tanh(\eta)$ and $\gamma = \mathrm{sigmoid}(\nu)$, and these notations then become $\boldsymbol{P}_k^{\mathrm{R}}(\eta)$, $\boldsymbol{P}_k^{\mathrm{C1}}(\nu, \theta)$ and $\boldsymbol{P}_k^{\mathrm{C2}}(\nu, \theta)$ with $k \in \{\mathrm{mask}, \mathrm{unmask}\}$ accordingly.

Sequential data usually have some recurrent patterns, which can be captured by REMs. In the meanwhile, the remaining non-recurrent patterns can be modeled by the baseline Transformer via $\mathrm{softmax}(\boldsymbol{Q}\boldsymbol{K}')$. The allocation of weights between REMs and self-attention is adjusted by the learnable gate $\sigma(\mu)$. On one hand, comparing with the baseline Transformer, the inclusion of REMs will lead to a significant improvement of sample efficiency in modeling the recurrent patterns, and hence a higher accuracy in prediction can be expected from the proposed RSA. On the other hand, comparing with the recurrent models, the non-recurrent patterns can be taken care of by the flexible Transformers, and hence the representation power of the RSA module is as good as that of the baseline Transformer.

In addition, the REM can act as a complementary positional encoding scheme to further provide relative location information. For the multihead RSA, the gate-control parameter $\mu$ only varies across layers, while the parameters controlling the matrix $\boldsymbol{P}$ vary across all heads and layers.

**Initialization.** The gate-control parameter $\mu$ is initialized in the interval $[-3, 3]$ for all the layers. For $(\lambda, \gamma, \theta)$ which determine the recurrent patterns, we initialize $\lambda$'s at different heads to spread out between $[-2, -1] \cup [1, 2]$ and $\nu$'s to spread out between $[1, 2]$, and $\theta$ is initialized at $\pi/4$, to encourage REMs to be non-zero and well-diversified.

**Dilated REM variants.** The dilated REMs can be further obtained by considering the block matrix formed by the first $T$ columns and first $T$ rows of $\boldsymbol{P} \otimes \boldsymbol{I}_d$, where $d$ is the dilating factor. In fact, the dilated REMs can encapsulate the recurrence patterns of the dilated RNNs (Chang et al., 2017); see Proposition 4 in the Appendix. They describe potentially periodic recurrence over a long-term timespan, and can significantly enrich the temporal dynamics of our RSA.

**Hyperparameters $k_i$ and $d$.** In each multihead RSA layer with $H$ heads, the number of the six types of REMs, namely one regular, two cyclical, one dilated regular and two dilated cyclical REMs, are denoted by $k_1 - k_6$, respectively. Since $\sum_{i=1}^{6} k_i = H$, we can apply constrained hyperparameter search to optimize over the choices of $k_i$'s. For simplicity, we can set $k_2 = k_3$ and $k_5 = k_6$, which indicates that the cyclical REMs come in pairs. The search for $d$, on the other hand, can be guided by various data analytic tools. For instance, $d$ can be observed from the autocorrelation plots as the

Figure 3: The boxplot for gate values at different attention layers of RSA models.

seasonal period length for time series data (Cryer & Chan, 2008); while for the language-related data, $d$ can be heuristically deduced from the recurrence plots (Webber et al., 2016).

**More discussions on REMs.** Each type of REMs is basically a masked or unmasked linear aggregation of tokens, and we may alternatively consider a more general Toeplitz, or even a fully learnable, matrix $\boldsymbol{P}$ at (4) such that all kinds of temporal decay patterns can be automatically captured. Although more flexible than REMs, it will need $O(T)$ or $O(T^2)$ additional parameters, where $T$ is the sequence length, while each REM requires merely one or two parameters. Note that the proposed RSA at (4) also includes a standard self-attention, which is flexible enough for the remaining nonlinearities that failed to be captured by REMs, and hence it may not be necessary to consider a more general yet less efficient structure for REMs. see Appendix B for both theoretical and empirical evidences.

## 4 EXPERIMENTS

This section contains four sequential modeling tasks and, for each task, we modify some popular Transformer baselines by adding the REMs to their attention weights via a gated mechanism as in (4). The modified models are named with the prefix "RSA-". The trained gate $\sigma(\mu)$ controls the percentage of weights allocated to the REMs, and it hence provides a rough measure for recurrent signals in the data. From Figure 3, we may argue that time series have the strongest recurrent signals, followed by the regular language (Yu, 1997) and finally the code or natural languages. The gate values also vary across different layers of Transformer models, indicating that the proportion of recurrent signals may change as the data propagate through layers. Moreover, the data-driven gated mechanism can help maintain the optimal allocation between REM and self-attention to achieve better sample efficiency and improved performance. All experiments are conducted on Nvidia V100 32GB GPUs.

### 4.1 TIME SERIES FORECASTING

Time series data possess unique attributes, such as strong recurrent signals, that can improve forecasting accuracy. Although Transformers are not currently the state-of-the-art model for this task, it is interesting to explore how the proposed RSA module could enhance their performance.

Our experiments are performed on two public benchmark datasets: the ETT[1] dataset is comprised of seven features related to the electric power long-term deployment, where $\{\text{ETTh}_1, \text{ETTh}_2\}$ are recorded by the hour and $\text{ETTm}_1$ is recorded by 15-minute intervals; and the Weather[2] dataset contains twelve climate indicators collected every 1 hour over a 4-year period. For the baseline models, we adopt Informer (Zhou et al., 2021), LogSparse Transformer (Li et al., 2019) and Transformer-XL (Dai et al., 2019). In particular, to accommodate the cases where the attention map is non-square, we extend the recurrent patterns and adjust the shape of our REMs; see details in Appendix D.2. All hyperparameters in baseline models are set to the optimal setting in Zhou et al. (2021), and we also follow their train/val/test division and training schemes to conduct our experiments.

Table 1 summarizes the mean squared error (MSE) and mean absolute error (MAE) for the three baselines against their RSA counterparts, averaged across 5 repetitions. Overall, the revised models show significant improvement in performance over their baselines. Moreover, it is interesting to see that the REMs can act as effective substitutions for positional encoding. For Transformer-XL in particular, although we remove their learnable relative positional encoding, superior performance is attained by RSA-XL while the total number of parameters is reduced. Since more than 70% of weights are allocated to the REMs in Figure 3, it numerically verifies that the recurrent signals dominate the non-recurrent ones for time series data.

---

[1] Accessible at `https://github.com/zhouhaoyi/ETDataset`.

[2] Accessible at `https://www.ncei.noaa.gov/data/local-climatological-data/`.

Table 1: Multivariate long sequence time-series forecasting results reported on different prediction window sizes. We compare the MSE and MAE of RSA models with their baselines under all settings, and highlight the better results in bold letters. Note that XL requires a sequential ordering for training, and the same scheme are adopted for all models to make the results comparable.

| Methods | | Informer | | RSA-Informer | | LogSparse | | RSA-LogSparse | | XL | | RSA-XL | |
|---|---|---|---|---|---|---|---|---|---|---|---|---|---|---|
| Metric | | MSE | MAE | MSE | MAE | MSE | MAE | MSE | MAE | MSE | MAE | MSE | MAE |
| ETTh1 | 24 | 0.762 | 0.632 | **0.414** | **0.450** | 1.124 | 0.826 | **0.858** | **0.668** | 0.514 | 0.518 | **0.466** | **0.493** |
| | 48 | 1.006 | 0.763 | **0.467** | **0.493** | 1.161 | 0.841 | **0.818** | **0.659** | 0.571 | 0.556 | **0.528** | **0.535** |
| | 168 | 1.141 | 0.823 | **0.753** | **0.659** | 1.104 | 0.818 | **1.042** | **0.781** | 0.898 | 0.734 | **0.813** | **0.694** |
| | 336 | 1.416 | 0.987 | **0.895** | **0.755** | 1.178 | 0.851 | **0.972** | **0.774** | 0.963 | 0.758 | **0.942** | **0.756** |
| ETTh2 | 24 | 2.558 | 1.253 | **1.264** | **0.879** | 2.894 | 1.375 | **1.076** | **0.844** | 0.763 | 0.699 | **0.705** | **0.660** |
| | 48 | 2.487 | 1.268 | **1.878** | **1.067** | 3.009 | 1.363 | **1.362** | **0.975** | 1.293 | 0.917 | **1.171** | **0.877** |
| | 168 | 2.869 | 1.324 | **2.830** | **1.301** | 2.876 | 1.307 | **2.165** | **1.235** | 2.780 | **1.288** | **2.671** | 1.290 |
| | 336 | **2.055** | **1.113** | 2.113 | 1.124 | 3.005 | **1.350** | **2.909** | 1.365 | **2.447** | **1.231** | 2.461 | 1.238 |
| ETTm1 | 24 | 0.536 | 0.511 | **0.534** | **0.507** | 1.105 | 0.837 | **0.619** | **0.553** | **0.561** | **0.537** | 0.591 | 0.549 |
| | 48 | 0.781 | 0.633 | **0.644** | **0.612** | 1.150 | 0.852 | **0.541** | **0.519** | 0.562 | 0.543 | **0.556** | **0.536** |
| | 96 | 0.823 | 0.697 | **0.732** | **0.665** | 1.227 | 0.897 | **0.526** | **0.520** | 0.714 | **0.640** | **0.707** | 0.645 |
| | 288 | 1.371 | 0.945 | **0.835** | **0.710** | 1.167 | 0.862 | **0.955** | **0.766** | 0.969 | 0.795 | **0.967** | **0.795** |
| Weather | 24 | **0.316** | **0.371** | 0.328 | 0.380 | 0.560 | 0.553 | **0.394** | **0.441** | **0.364** | **0.411** | 0.367 | 0.414 |
| | 48 | 0.606 | 0.566 | **0.432** | **0.464** | 0.582 | 0.567 | **0.432** | **0.466** | 0.473 | 0.490 | **0.466** | **0.484** |
| | 168 | 1.009 | 0.771 | **0.862** | **0.702** | 0.929 | 0.754 | **0.602** | **0.580** | 0.684 | 0.616 | **0.592** | **0.572** |
| | 336 | 1.096 | 0.801 | **0.846** | **0.697** | 0.874 | 0.734 | **0.638** | **0.602** | 0.895 | 0.713 | **0.816** | **0.679** |

Table 2: Prediction accuracy of the vanilla Transformer, the Transformer with learnable relative positional encoding ("Transformer*") on the six languages, together with the RSA-Transformer under different hyperparameter settings. All Transformers are 5-headed, and the hyperparameter settings for RSA-Transformer, Cases I-IV, are denoted by the tuple $(k_1, k_2, k_3, k_4, k_5, k_6)$.

| Methods | Transformer | | Transformer* | | RSA-Transformer | | | | | | | |
|---|---|---|---|---|---|---|---|---|---|---|---|---|
| | | | | | I: (5,0,0,0,0,0) | | II: (3,0,0,2,0,0) | | III: (3,1,1,0,0,0) | | IV: (3,0,0,0,1,1) | |
| | Bin 0 | Bin 1 | Bin 0 | Bin 1 | Bin 0 | Bin 1 | Bin 0 | Bin 1 | Bin 0 | Bin 1 | Bin 0 | Bin 1 |
| $D_2$ | 0.2 | 0.2 | 0.87 | 0.94 | **1** | **1** | **1** | **1** | **1** | **1** | **1** | **1** |
| $D_4$ | **1** | 0.08 | 0.99 | 0.9 | **1** | **1** | **1** | **1** | **1** | **1** | **1** | **1** |
| Parity | 0.29 | 0 | 0.13 | 0 | **0.99** | **0.67** | 0.97 | 0.53 | 0.91 | 0.62 | 0.9 | 0.52 |
| Tomita 3 (T3) | 0.89 | 0.11 | 0.98 | 0.48 | **1** | 0.97 | **1** | 0.97 | **1** | **0.98** | **1** | **0.98** |
| Tomita 5 (T5) | 0.07 | 0 | 0.01 | 0 | 0.63 | 0.16 | **0.82** | 0.17 | 0.49 | 0 | 0.72 | **0.35** |
| Tomita 6 (T6) | 0 | 0 | 0 | 0 | 0.78 | 0.35 | 0.89 | 0.38 | **0.95** | **0.46** | 0.64 | 0.39 |

## 4.2 REGULAR LANGUAGE LEARNING

Regular languages are intimately related to the linear recurrence sequences, such as Fibonacci numbers (Kotek & Makowsky, 2010). Some works report that Transformers have difficulty in generalizing the rules of regular languages (Bhattamishra et al., 2020; Tran et al., 2018; Hahn, 2020). Experiments are conducted to show that the proposed RSA module can leverage the recurrent inductive bias of the REMs to improve Transformer's sample efficiency, and subsequently its generalization ability.

For the experiments, we deliberately choose six types of regular languages on which the Transformer model has been shown to perform poorly, namely Parity, Tomita 3, 5 & 6 (Tomita, 1982), $D_2$ and $D_4$ (Bhattamishra et al., 2020). We use the decoder-only vanilla Transformer model as the baseline, and modify its self-attention layers into our RSA forms under four hyperparameter settings (detailed in Table 2). As a comparison, another benchmark model (denoted by Transformer*) is obtained by replacing the absolute sinusoidal positional encoding with the learnable relative positional encoding (Dai et al., 2019). To evaluate their generalization ability, the test set is further separated into two bins, where bin 1 contains longer test samples than bin 0. Following Bhattamishra et al. (2020), we recast the prediction task into a classification one, where the prediction is deemed a success only when the output values at all positions are correct, and a failure otherwise.

From Table 2, it can be seen that the vanilla Transformer model fails to grasp the recurring rules of these languages. With assistance of the learned relative positional encoding, they can learn better on $D_2, D_4$ and Tomita 3, but not the others. However, the RSA-Transformer not only achieves near 100% accuracy on $D_2, D_4$ and Tomita 3, but also gains a significant boost in performance on the other three languages. In particular, Figure 3 shows that approximately 60% of weights are allocated to the REMs, implying strong recurrent signals in the datasets. We further provide some visualization of the REMs under Cases I – IV and their interpretaions in Appendix E.2.

Table 3: (a) The defect accuracy is reported for the CodeT5-small baseline and the RSA-CodeT5-small. In addition, we report the percentage of the parameters added by the RSA model against its baseline. (b) The number of Bit-per-character (BPC) is reported for Enwik8 and Text8 (lower is better), and the perplexity (PPL) is reported for WikiText-103 (lower is better).

| Models | Defect Accuracy (%) |
|---|---|
| CodeT5-small | 64.60 |
| RSA-CodeT5-small | **65.96** |
| # Params Added (%) | $5 \times 10^{-4}$ |

(a) Code defect detection

| Models | XL | RSA-XL | BRT | RSA-BRT |
|---|---|---|---|---|
| Enwik8 | 1.074 | **1.068** | 1.076 | **1.068** |
| Text8 | 1.163 | 1.163 | 1.165 | **1.163** |
| WikiText-103 | 23.83 | **23.73** | 23.76 | **23.64** |
| # Averaged Params Added (%) | $1.01 \times 10^{-4}$ | | $8.68 \times 10^{-5}$ | |

(b) Natural language modeling

### 4.3 CODE AND NATURAL LANGUAGE MODELING

Different from regular languages, the recurrence relationship in programming or natural languages is weaker and harder to interpret. Nevertheless, they have a wide range of applications on which the Transformers are serving as the baseline models. Our experiments show that the baseline models can improve the performance with inclusion of a small percentage of REMs.

We first conduct a defect detection task based on the C language dataset provided by Zhou et al. (2019), which is a binary classification task to evaluate whether a code is vulnerable to external attacks. The pretrained CodeT5 (Wang et al., 2021)[3] is adapted to this downstream task. Specifically, we use CodeT5-small as our baseline model with 12 layers and 8 heads, and directly add the REMs on top of the self-attention with fully learnable RPE via a gated mechanism. The gate-control parameter $\mu$ is initialized at $-3$ and we fine-tune both the baseline and our RSA models for 20 epochs with early stopping. It can be observed from the boxplot in Figure 3 that the gate values at different layers remain close to the initialization, with only 5% of weights allocated to the REMs. Although the recurrent signals in the code language is apparently much weaker than those in time series or regular language, our RSA-CodeT5-small is still able to achieve a significant improvement against the CodeT5-small baseline as shown in Table 3(a). The RSA-CodeT5-small model has only about 320 additional parameters over the baseline with 66 million parameters, which is equal to $5 \times 10^{-4}\%$. More surprisingly, it surpasses the current SOTA result of 65.78%, which is achieved by the CodeT5-base model with a total number of 220 million parameters (Wang et al., 2021). Due to limited resources, we have not yet fine-tuned RSA-CodeT5-base model.

The natural language modeling task is conducted on three standard NLP datasets, namely the word-level WikiText-103 (Merity et al., 2017), Enwik8 (LLC., 2009) and Text8 (LLC., 2009). We use two baseline models: Transformer-XL and Block Recurrent Trasnformer (BRT), and their corresponding RSA models are formed by directly adding the REMs on top of some of the self-attention heads via a gated mechanism. For Wikitext-103, all transformers are 16-layer with 12 heads. In the meantime, for both Enwik8 and Text8 there are 14 layers and 8 heads. The details of model structures can be referred to Appendix E.3. We follow the training schemes of Transformer-XL, and the results are reported in Table 3(b). Overall, it can be seen that RSA models can achieve better performance than their corresponding baselines. The gated values for RSA-XL on the three datasets are presented in Figure 3, and we can see that REMs contribute more substantially for Text8 and Enwik8 than WikiText-103, which may imply stronger recurrent signals for these two datasets.

### 4.4 ABLATION STUDIES

This section conducts three experiments to analyze how RSA scales with respect to sample size, data size and sequence length, and its performance is compared against the baseline models.

**Model scaling** The first ablation studies the performance of RSA-XL on Text8 dataset when the number of parameters increases. Baseline models include Transformer-XL, and two other benchmarks suggested by a reviewer, which replace the parsimonious matrix function $P$ in RSA (see equation (4)) with (1) a Toeplitz matrix, or (2) a fully learnable matrix. They are incorporated into Transformer-XL baseline and are hence referred to as TL-XL and FL-XL, respectively. To have comparable parameter counts, the number of layers in FL-XL varies from 8 to 14, whereas that in others varies from 10 to 16. We adopt the original training schemes of Transformer-XL. From the results presented in Figure 4 (left), it can be observed that RSA-XL is the most parameter efficient across all model scales. See Appendix E.4 for similar results comparing BRT and RSA-BRT on model scaling performance.

---

[3]Pretrained weights available from `https://github.com/salesforce/CodeT5`.

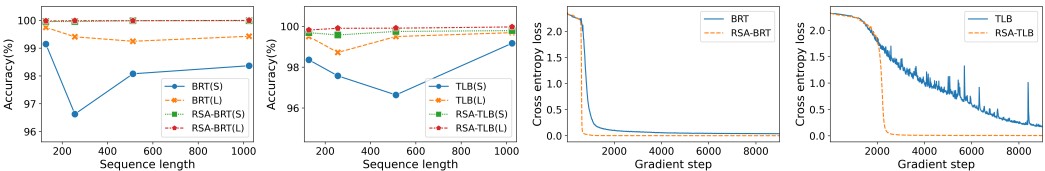

Figure 4: The left panel presents the BPC on Text8 for XL, RSA-XL, TL-XL and FL-XL at different model sizes. The middle panel compares the sample efficiency of RSA-XL on with that of the baseline XL by training them on restricted portions of the Enwik8 training set. Meanwhile, the gate-values at different attention layers of RSA-XL are presented in the boxplot.

Figure 5: Accuracy for BRT vs RSA-BRT and TLB vs RSA-TLB on the copy task is plotted on the first and second panels respectively, where "S" means a small training sample size of 6400, and "L" means a large training sample size of 12800. When sequence length = 512, the cross entropy loss of the four models for the first 9000 gradient steps are plotted in the third and fourth panels.

**Data scaling** The second ablation study compares the sample efficiency of RSA-XL against Transformer-XL. Specifically, we use different proportions of training data from Enwik8 to train the models, and maintain the same evaluation and test sets. Results are presented in middle panel of Figure 4, and we further visualize the gate values $\sigma(\nu)$ allocated to REMs at every layer in the right panel. It shows that, as available training samples shrink, learnable gate will automatically increase the weights of the parsimonious REMs, and this subsequently helps RSA-XL to maintain a consistently higher sample efficiency than the baseline XL model.

**Sequence length scaling** In this ablation study, we adopt a synthetic copy task akin to the one in Nguyen et al. (2021) or Katharopoulos et al. (2020). It requires the model to copy a sequence of symbols with different maximum lengths. For every sample, a sequence of ten symbols is randomly generated and repeated multiple times, with a reserved symbol separating each replication. Four models are used: BRT, TLB, RSA-BRT and RSA-TLB, and we adopt a small training sample size of 6400 as well as a larger one of 12800. The implementation is detailed in the appendix. Figure 5 first depicts the accuracy with respect to the sequence length and then shows the cross entropy loss with respect to the number of gradient steps. It can be observed that RSA-BRT and RSA-TLB converge faster than their baselines, respectively, and achieve perfect accuracy across all sequence lengths.

## 5 CONCLUSION AND DISCUSSION

By formulating RNN into an MHSA form, we novelly propose an RSA module to incorporate the recurrent dynamics of an RNN into a Transformer. Specifically, the lightweight REMs are combined with the self-attention weights via a gated mechanism, maintaining a parallel and efficient computation on the GPU infrastructure. Experiments on four sequential learning tasks show that the proposed RSA module can boost the sample efficiency of the baseline Transformer, supported by significantly improved performance. In addition, with the fitted gate values to measure the recurrent signal strength in the data, it can be observed that the code or natural languages have much smaller proportion of recurrent patterns, which further explain the superiority of Transformer models in these two data domains, while the inclusion of REMs still can boost the baseline performance.

This paper can be extended along three directions below. Firstly, while six typical examples of REMs are introduced, we may also consider other types of REM designs, such as the non-distinct eigenvalues of the RNN weight matrix $\boldsymbol{W}_h$, and long-memory filters operated on top of the RNNs (Zhao et al., 2020). Secondly, the proposed RSA module can be further applied to other types of sequential learning tasks such as video learning (Bulat et al., 2021) and skeleton-based human motion detection (Shu et al., 2021). Finally, it is of interest to incorporate the REMs into models other than Transformers, or even to apply it directly as a standalone recurrent unit.

## 6 REPRODUCIBILITY STATEMENT

To ensure the reproducibility and completeness of this paper, we include the Appendix with five main sections. Appendix A discusses the relationship between RSA and some important related works. Appendix B provides a theoretical and empirical study on the performance gap between linear and nonlinear RNNs. Appendix C contains the complete proofs for all propositions presented in this paper. An efficient method to compute the REMs is detailed in Appendix D, together with an extension to non-square REMs and an empirical study on the time efficiency of the proposed method. The experiments in the paper are reproducible with additional implementation details provided in Appendix E. We also include the hyperparameter settings for all results contained in Tables 1 and 3.

### ACKNOWLEDGMENTS

We would like to thank the three anonymous reviewers for spending time and efforts and bringing in constructive questions and suggestions, which help us greatly to improve the quality of the paper. We would like to also thank the Program Chairs and Area Chairs for handling this paper and providing the valuable and conprehensive comments.

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

APPENDIX

This Appendix contains five sections. The first section provides an illustrative plot for multiscale recurrence in various Transformer models. The theoretical and empirical gaps between the linear and nonlinear RNNs are studied in the second section. The third section presents Proposition 3, and then gives the detailed proofs for Propositions 1 – 3. An efficient method to compute the REMs is detailed in Section 4, and we further extend it to non-square REMs and conduct an empirical study on the training time per batch for the RSA models against their baselines. Section 5 provides additional implementation details for the experiments, additional ablation studies and visualization of the learned patterns.

## A    MULTISCALE RECURRENCE

For sequential learning tasks, the recurrence relationship can be well observed at different levels of temporal granularity. This feature has inspired new designs to add recurrence to Transformers from varying scales; see illustration in Figure 6. Transformer-XL (XL) (Dai et al., 2019) partitions a long input sequence into segments, and places them into consecutive batches. A segment-level recurrence is then introduced by using a cached hidden state to pass historical information iteratively to the next segment. Temporal Latent Bottleneck (TLB) (Didolkar et al., 2022) further divides the segment within one batch into smaller chunks, and then adopts the state vectors to aggregate both high-level information across layers and temporal information across chunks. The chunk-level recurrence helps create a slow information stream in the sequence to learn a more condensed representation. Block-Recurrent Transformer (BRT) (Hutchins et al., 2022) also establishes recurrence across chunks (or blocks), while their recurrent states are layer-specific and updated with an LSTM-style gated design. As a comparison, the proposed RSA follows the RNN to account for the recurrence between individual inputs. In other words, it models the token-level recurrence, which is at the most fine-grained scale. Subsequently, it can be easily incorporated into the aforementioned coarser-grained designs, and may potentially bring benefits to their performance. For illustration, we use XL and BRT as our baseline models in Section 4.3.

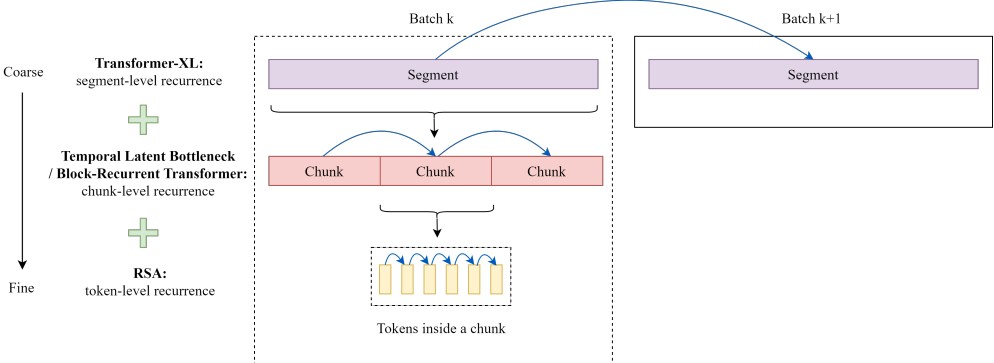

Figure 6: An illustration for multiscale recurrence. From top to bottom, Transformer-XL has the most coarse-grained segment-level recurrence, while both the Temporal Latent Bottleneck and the Block-Recurrent Transformers share the finer-grained chunk-level (or block-level) recurrence. The proposed RSA has the most fined-grained token-level recurrence.

## B    THEORETICAL AND EMPIRICAL GAPS BETWEEN LINEAR AND NONLINEAR RNNS

### B.1    THEORETICAL GAP

This subsection theoretically evaluates prediction errors when a linear RNN model is used to train the data generated by a nonlinear RNN. For simplicity, we consider 1D case only, and many-to-many RNNs are assumed.

Specifically, the nonlinear RNN model used to generate the data has the recursive form of

$$g_t = \sigma_h(u_t) \quad \text{with} \ \ u_t = w_h^* g_{t-1} + w_x^* x_t + b^*, \tag{5}$$

and $\sigma_h(\cdot)$ is the activation function satisfying

$$|\sigma_h(0)| < 1, \quad \sigma_h'(0) = 1 \quad \text{and} \quad |\sigma_h''(x)| \le 1 \text{ for any } x \in \mathbb{R}. \tag{6}$$

Note that many commonly used activation functions, including Tanh and Sigmoid, satisfy the above condition. We further consider an additive error $\varepsilon_t$, i.e. $y_t = g_t + \varepsilon_t$, where $\varepsilon_t$ has mean zero and a finite variance denoted by $\gamma$.

For the generated data $\{y_t\}$, we train a linear RNN model,

$$h_t(\boldsymbol{\theta}) = w_h h_{t-1}(\boldsymbol{\theta}) + w_x x_t + b,$$

where the parameters $\boldsymbol{\theta} = (w_h, w_x, b)$. Then the mean squared prediction error can be defined as

$$e_{\text{pred}} := \min_{\boldsymbol{\theta}} \mathbb{E}(y_t - h_t(\boldsymbol{\theta}))^2,$$

and its theoretical bound is provided in the following proposition.

**Proposition 3.** *Suppose that $\mathbb{E}(u_t^2) \le \alpha$ and $\mathbb{E}(u_t^2 u_s^2) \le \beta$ for all $t, s \in \mathbb{Z}$. If $|w_h^*| < 1$ and the condition at (6) holds, then*

$$e_{\text{pred}} \le \underbrace{(1 - |w_h^*|)^{-1}(1 + \alpha + \beta/4)}_{\text{misspecification error}} + \underbrace{\gamma}_{\text{irreducible system error}},$$

*where the first part is due to the misspecification of using the linear activation to approximate $\sigma_h(\cdot)$.*

*Proof of Proposition 3.* Let $\boldsymbol{\theta}^* = (w_h^*, w_x^*, b^*)$, and denote $h_t = h_t(\boldsymbol{\theta}^*)$ for all $t \in \mathbb{Z}$, i.e.

$$h_t = w_h^* h_{t-1} + w_x^* x_t + b^*. \tag{7}$$

By the definition of $e_{\text{pred}}$, it holds that

$$e_{\text{pred}} \le \mathbb{E}(y_t - h_t)^2 = \mathbb{E}(g_t - h_t)^2 + \gamma, \tag{8}$$

where the equality comes from $\mathbb{E}(\varepsilon_t) = 0$ and $\text{var}(\varepsilon_t) = \gamma$. By applying second-order Taylor expansion at zero and from (5) and (6), we have

$$g_t = \sigma_h(0) + u_t + R_t(0), \quad \text{where } |R_t(0)| = \left| \frac{1}{2}\sigma_h''(\widetilde{u})u_t^2 \right| \le \frac{u_t^2}{2},$$

and $\widetilde{u}$ lies between $u_t$ and zero. This, together with (7), leads to $g_t - h_t = \sigma_h(0) + w_h^*(g_{t-1} - h_{t-1}) + R_t(0)$.

Let $\delta_t = g_t - h_t$, and it then holds that

$$\delta_t = \sigma_h(0) + w_h^* \delta_{t-1} + R_t(0) = \sum_{j=0}^{\infty}(w_h^*)^j \sigma_h(0) + \sum_{j=0}^{\infty}(w_h^*)^j R_{t-j}(0),$$

where the second equality is obtained by applying the first equality recursively. As a result, by the condition that $|\sigma_h(0)| < 1$, $\mathbb{E}(u_t^2) \le \alpha$ and $\mathbb{E}(u_t^2 u_s^2) \le \beta$ for all $t, s \in \mathbb{Z}$, we can show that

$$\mathbb{E}(\delta_t^2) \le \xi^2 + 2\xi \mathbb{E}\left(\sum_{j=0}^{\infty}|w_h^*|^j|R_{t-j}(0)|\right) + \mathbb{E}\left(\sum_{j=0}^{\infty}|w_h^*|^j|R_{t-j}(0)|\right)^2$$

$$\le \xi^2\left(1 + \alpha + \frac{\beta}{4}\right),$$

where $\xi = \sum_{j=0}^{\infty}|w_h^*|^j$. If $|w_h^*| < 1$, we have $\xi = (1 - |w_h^*|)^{-1}$. This, together with (8), accomplishes the proof. □

## B.2 EMPIRICAL GAP

This subsection conducts a synthetic experiment to evaluate the performance of linear RNNs when there is nonlinearity in the data.

We first generate the data by using a two-layer nonlinear RNN model with the form of

$$\boldsymbol{z}_t^{(i)} = \alpha \boldsymbol{g}_t^{(i)} + (1-\alpha)\boldsymbol{h}_t^{(i)} \quad \text{and} \quad \begin{cases} \boldsymbol{h}_t^{(i)} = \boldsymbol{W}_h^{(i)}\boldsymbol{z}_{t-1}^{(i)} + \boldsymbol{W}_z^{(i)}\boldsymbol{z}_t^{(i-1)} + \boldsymbol{b}^{(i)} \\ \boldsymbol{g}_t^{(i)} = \sigma_h(\boldsymbol{W}_h^{(i)}\boldsymbol{z}_{t-1}^{(i)} + \boldsymbol{W}_z^{(i)}\boldsymbol{z}_t^{(i-1)} + \boldsymbol{b}^{(i)}) \end{cases}, \tag{9}$$

with $i = 1$ and $2$, where $\boldsymbol{z}_t^{(0)} = \boldsymbol{x}_t$, $\boldsymbol{z}_t^{(i)} \in \mathbb{R}^2$ for $1 \le i \le 3$, $\sigma_h(\cdot)$ is a nonlinear activation function, and $0 \le \alpha \le 1$ is the weight of nonlinearity. An additive error is further assumed, i.e. $\boldsymbol{y}_t = \boldsymbol{z}_t^{(2)} + \boldsymbol{\varepsilon}_t$,

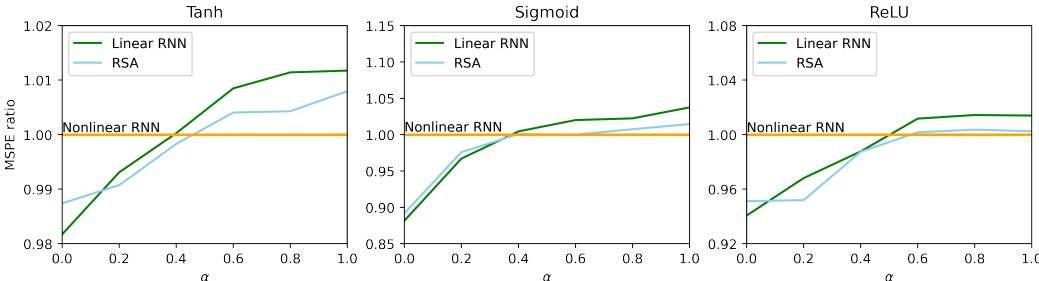

Figure 7: The MSPE ratios for the linear RNN and RSA as the nonlinearity proportion $\alpha$ changes from 0 to 1. From the left, middle to right panels, we set $\sigma_h(\cdot)$ in (9) to Tanh, Sigmoid or ReLU activation, respectively.

where $\{\boldsymbol{x}_t\}$ and $\{\boldsymbol{\varepsilon}_t\}$ are independent and follow the standard multivariate normal distribution. Three nonlinear functions are considered for $\sigma_h(\cdot)$, Tanh, Sigmoid and ReLU. As $\alpha$ increases from 0 to 1, the data generating process gradually changes from a strictly linear RNN to a nonlinear one, i.e. $\alpha$ essentially controls the proportion of nonlinearity involved.

The sequence $\{\boldsymbol{y}_t, 1 \leq t \leq T\}$ is fitted separately by a linear RNN, a nonlinear RNN with the corresponding activation, and a linear RNN combined with a self-attention, i.e. the proposed RSA. Specifically, we generate a sequence of length 10000 and then divide it into 100 segments, each of length 100. In each segment, we train with the first 99 observations and calculate the prediction error for the last observation. The Adam optimizer is adopted for training, and the training procedure will be terminated when the training loss drops by less than $10^{-5}$. The mean squared prediction error (MSPE) averaged over 100 segments are denoted by $e_{\text{pred}}^{\text{L}}, e_{\text{pred}}^{\text{NL}}$ and $e_{\text{pred}}^{\text{RSA}}$ for the three models, respectively. Using nonlinear RNNs as the benchmark, the MSPE ratio for the linear RNN or the RSA is defined as

$$\text{MSPE ratio for model } i = \frac{e_{\text{pred}}^i}{e_{\text{pred}}^{\text{NL}}}, \quad \text{where } i \in \{\text{L}, \text{RSA}\}.$$

Figure 7 presents the MSPE ratios for three types of activation functions. It can be seen that, when $\alpha = 1$, nonlinear RNNs perform the best, while linear RNN suffers from misspecification error. Alternatively, when $\alpha = 0$, the opposite can be observed. Moreover, as $\alpha$ increases, i.e. there are more nonlinearity, it is expected that linear RNNs become less favorable, while the proposed RSA can remedy the problem to some extent. Especially when $\alpha > 0.6$, the RSA consistently achieves better prediction performance than the pure linear RNN.

## C  PROPOSITION 3 AND PROOFS FOR ALL PROPOSITIONS

### C.1  PROPOSITION 3 FOR DILATED RNNS

For some positive integer $d$, let $\bar{\boldsymbol{P}}_{\text{mask}}^{\text{R}}$ be the block matrix formed by the first $T$ columns and the first $T$ rows of $\boldsymbol{P}_{\text{mask}}^{\text{R}} \otimes \boldsymbol{I}_d$. And $\bar{\boldsymbol{P}}_{\text{mask}}^{\text{C}_i}$ can be defined similarly for $i = 1$ or $2$. Consider a dilated RNN (Chang et al., 2017) with the dilating factor $d$. It has the form $\boldsymbol{h}_t = g(\boldsymbol{W}_h \boldsymbol{h}_{t-d} + \boldsymbol{W}_x \boldsymbol{x}_t + \boldsymbol{b})$ where $g(\cdot)$ is the activation function, $\boldsymbol{h}_t \in \mathbb{R}^d$ is the output or hidden variable with $\boldsymbol{h}_0 = 0$, $\boldsymbol{b} \in \mathbb{R}^{d_{\text{in}}}$ is the bias term, $\boldsymbol{W}_h \in \mathbb{R}^{d \times d}$ and $\boldsymbol{W}_x \in \mathbb{R}^{d \times d_{\text{in}}}$ are weights. When the activation function is linear, i.e. $g(x) = x$, the RNN becomes

$$\boldsymbol{h}_t = \boldsymbol{W}_h \boldsymbol{h}_{t-d} + \boldsymbol{W}_x \boldsymbol{x}_t, \quad \text{or} \quad \boldsymbol{h}_t = \sum_{j=0}^{t-1} \boldsymbol{W}_h^j \boldsymbol{W}_x \boldsymbol{x}_{t-j \cdot d}, \tag{10}$$

where the bias term $\boldsymbol{b}$ is suppressed for simplicity. We have the following proposition.

**Proposition 4.** *If the conditions of Proposition 1 hold for $\boldsymbol{W}_h$ in (10), then the RNN with linear activation at (10) can be represented into a multihead self-attention (MHSA) with $r + 2s + 1$ heads, where the query and key matrices are zero, and relative positional encoding matrices are $\{\bar{\boldsymbol{P}}_{\text{mask}}^{\text{R}}(\lambda_k), 1 \leq k \leq r\}$, $\{\check{\boldsymbol{P}}_{\text{mask}}^{\text{C}1}(\gamma_k, \theta_k), \bar{\boldsymbol{P}}_{\text{mask}}^{\text{C}2}(\gamma_k, \theta_k), 1 \leq k \leq s\}$ and an identity matrix, respectively.*

*Proof.* The proof follows directly from Propositions 1 and 2.  □

## C.2 PROOF FOR PROPOSITION 1

Let $\boldsymbol{W}_h$ be a $d \times d$ real matrix with distinct eigenvalues, and from Chapter 3 of Horn & Johnson (2012), we have the Jordan decomposition, $\boldsymbol{W}_h = \boldsymbol{B}\boldsymbol{J}\boldsymbol{B}^{-1}$, where $\boldsymbol{B} \in \mathbb{R}^{d \times d}$ is invertible, and $\boldsymbol{J} \in \mathbb{R}^{d \times d}$ has a real Jordan form, $\boldsymbol{J} = \operatorname{diag}\{\lambda_1, \ldots, \lambda_r, \boldsymbol{C}_1, \ldots, \boldsymbol{C}_s, \boldsymbol{0}\}$ with

$$\boldsymbol{C}_k = \gamma_k \cdot \begin{pmatrix} \cos(\theta_k) & \sin(\theta_k) \\ -\sin(\theta_k) & \cos(\theta_k) \end{pmatrix} \in \mathbb{R}^{2 \times 2}, \quad 1 \le k \le s.$$

Then,

$$\boldsymbol{W}_h^j = \boldsymbol{B}\boldsymbol{J}^j\boldsymbol{B}^{-1} = \sum_{k=1}^{r} \lambda_k^j \boldsymbol{G}_k^{\mathrm{R}} + \sum_{k=1}^{s} \gamma_k^j \left\{ \cos(j\theta_k)\boldsymbol{G}_k^{\mathrm{C1}} + \sin(j\theta_k)\boldsymbol{G}_k^{\mathrm{C2}} \right\} \quad \text{for all } j \ge 1,$$

where $\boldsymbol{G}_k^{\mathrm{R}}$'s, $\boldsymbol{G}_k^{\mathrm{C1}}$'s and $\boldsymbol{G}_k^{\mathrm{C2}}$'s are $d \times d$ real matrices determined jointly by $\boldsymbol{B}$ and $\boldsymbol{B}^{-1}$. Let $\boldsymbol{h}_{0,t} = \boldsymbol{W}_x \boldsymbol{x}_t$ and then,

$$\begin{aligned}
\boldsymbol{h}_t &= \sum_{j=0}^{t-1} \boldsymbol{W}_h^j \boldsymbol{W}_x \boldsymbol{x}_{t-j} \\
&= \sum_{j=1}^{t-1}\sum_{k=1}^{r} \lambda_k^j \boldsymbol{G}_k^{\mathrm{R}} \boldsymbol{W}_x \boldsymbol{x}_{t-j} + \sum_{j=1}^{t-1}\sum_{k=1}^{s} \gamma_k^j \left\{ \cos(j\theta_k)\boldsymbol{G}_k^{\mathrm{C1}} + \sin(j\theta_k)\boldsymbol{G}_k^{\mathrm{C2}} \right\} \boldsymbol{W}_x \boldsymbol{x}_{t-j} + \boldsymbol{W}_x \boldsymbol{x}_t \\
&= \sum_{k=1}^{r} \boldsymbol{h}_t^{\mathrm{R}}(\lambda_k) + \sum_{k=1}^{s} \boldsymbol{h}_t^{\mathrm{C1}}(\gamma_k, \theta_k) + \sum_{k=1}^{s} \boldsymbol{h}_t^{\mathrm{C2}}(\gamma_k, \theta_k) + \boldsymbol{h}_{0,t},
\end{aligned}$$

where

$$\boldsymbol{h}_t^{\mathrm{R}} = \sum_{j=1}^{t-1} \lambda^j \boldsymbol{G}^{\mathrm{R}} \boldsymbol{W}_x \boldsymbol{x}_{t-j} \quad \text{or equivalently,} \quad \boldsymbol{h}_t^{\mathrm{R}} = g(\lambda \boldsymbol{h}_{t-1}^{\mathrm{R}} + \boldsymbol{G}^{\mathrm{R}} \boldsymbol{W}_x \boldsymbol{x}_{t-1}), \tag{11}$$

$g(\cdot)$ being the identity function and $\boldsymbol{h}_t^{\mathrm{C1}} = \sum_{j=1}^{t-1} \gamma^j \cos(j\theta)\boldsymbol{G}^{\mathrm{C1}}\boldsymbol{W}_x \boldsymbol{x}_{t-j}$, $\boldsymbol{h}_t^{\mathrm{C2}} = \sum_{j=1}^{t-1} \gamma^j \sin(j\theta)\boldsymbol{G}^{\mathrm{C2}}\boldsymbol{W}_x \boldsymbol{x}_{t-j}$ which can be obtained via the recursive relationships,

$$\begin{cases} \boldsymbol{h}_t^{\mathrm{C1}} = g(\gamma\cos\theta\boldsymbol{h}_{t-1}^{\mathrm{C1}} + (\gamma\cos\theta\boldsymbol{G}\boldsymbol{W}_x\boldsymbol{x}_{t-1} - \gamma\sin\theta\boldsymbol{h}_{t-1}^{\mathrm{C2}})), \\ \boldsymbol{h}_t^{\mathrm{C2}} = g(\gamma\cos\theta\boldsymbol{h}_{t-1}^{\mathrm{C2}} + \gamma\sin\theta(\boldsymbol{h}_{t-1}^{\mathrm{C1}} + \boldsymbol{G}\boldsymbol{W}_x\boldsymbol{x}_{t-1})), \end{cases} \tag{12}$$

with $\boldsymbol{G}$ being $\boldsymbol{G}^{\mathrm{C1}}$ or $\boldsymbol{G}^{\mathrm{C2}}$, respectively. For a more general form, we can further assume that the activation function $g(\cdot)$ is nonlinear for the simple RNNs at (11) and (12).

## C.3 PROOF FOR PROPOSITION 2

Using the SA form, we can represent the three types of the RNNs by

$$(\boldsymbol{h}_1^{\mathrm{R}}(\lambda_k), \cdots, \boldsymbol{h}_1^{\mathrm{R}}(\lambda_k)) = \mathrm{SA}_k^{\mathrm{R}}(\boldsymbol{X}) \text{ and}$$

$$(\boldsymbol{h}_1^{\mathrm{C}i}(\gamma_k, \theta_k), \cdots, \boldsymbol{h}_1^{\mathrm{C}i}(\gamma_k, \theta_k)) = \mathrm{SA}_k^{\mathrm{C}i}(\boldsymbol{X}) \text{ for } i = 1 \text{ or } 2.$$

Therefore, the first term in Proposition 1 can be represented as

$$\mathrm{MHSA}(\boldsymbol{X}) = \mathrm{concat}[\mathrm{SA}_1^{\mathrm{R}}(\boldsymbol{X}), \ldots, \mathrm{SA}_r^{\mathrm{R}}(\boldsymbol{X})]\boldsymbol{W}_o = \left( \sum_{k=1}^{r} \boldsymbol{h}_1^{\mathrm{R}}(\lambda_k), \ldots, \sum_{k=1}^{r} \boldsymbol{h}_T^{\mathrm{R}}(\lambda_k) \right)', \tag{13}$$

where $\boldsymbol{W}_o = (\boldsymbol{I}_d, \ldots, \boldsymbol{I}_d)' \in \mathbb{R}^{rd \times d}$ with $\boldsymbol{I}_d$ being $d$-dimensional identity matrix. Similarly the MHSA for second and third term in Proposition 1 is given by

$$\mathrm{MHSA}(\boldsymbol{X}) = \mathrm{concat}[\mathrm{SA}_1^{\mathrm{C}i}(\boldsymbol{X}), \ldots, \mathrm{SA}_s^{\mathrm{C}i}(\boldsymbol{X})]\boldsymbol{W}_o = \left( \sum_{k=1}^{s} \boldsymbol{h}_1^{\mathrm{C}i}(\gamma_k, \theta_k), \ldots, \sum_{k=1}^{s} \boldsymbol{h}_T^{\mathrm{C}i}(\gamma_k, \theta_k) \right)'$$

$$\tag{14}$$

where $\boldsymbol{W}_o = (\boldsymbol{I}_d, \ldots, \boldsymbol{I}_d)' \in \mathbb{R}^{sd \times d}$.

And we define the additional head as $\mathrm{SA}_0(\boldsymbol{X}) = (\boldsymbol{h}_{0,1}, \ldots, \boldsymbol{h}_{0,T})' = [\mathrm{softmax}(\boldsymbol{Q}\boldsymbol{K}') + \boldsymbol{I}]\boldsymbol{V}$ with $\boldsymbol{W}_Q = \boldsymbol{W}_K = 0$ and $\boldsymbol{W}_V = \boldsymbol{W}_x'$. Combine with (13) and (14), we have

$$\text{MHSA}(\boldsymbol{X}) = \text{concat}[\{\text{SA}_k^{\text{R}}(\boldsymbol{X})\}_{1 \le k \le r}, \{\text{SA}_k^{\text{C1}}(\boldsymbol{X}), \text{SA}_k^{\text{C2}}(\boldsymbol{X})\}_{1 \le k \le s}, \text{SA}_0(\boldsymbol{X})]\boldsymbol{W}_o \quad (15)$$

where $\boldsymbol{W}_o = (\boldsymbol{I}_d, \ldots, \boldsymbol{I}_d)' \in \mathbb{R}^{(r+2s+1)d \times d}$.

## D  COMPUTATION OF THE REMS

### D.1  EFFICIENT COMPUTATION OF THE REMS

For any matrix $\boldsymbol{A}$, let $\cos(\boldsymbol{A})$ and $\sin(\boldsymbol{A})$ refer to applying the cosine and sine functions to each element of $\boldsymbol{A}$. To compute the REMs, we first construct the exponent matrix $\boldsymbol{L}$ that has a Toeplitz form:

$$\boldsymbol{L} = \begin{pmatrix} 0 & 1 & 2 & \cdots & T-1 \\ 1 & 0 & 1 & \cdots & T-2 \\ 2 & 1 & 0 & \cdots & T-3 \\ \vdots & \vdots & \vdots & \ddots & \vdots \\ T-1 & T-2 & T-3 & \cdots & 0 \end{pmatrix}$$

Denote the square matrix whose elements are all equal to $\lambda$ as $\boldsymbol{\lambda}$. Similarly, we define $\boldsymbol{\gamma}$ and $\boldsymbol{\theta}$ for $\gamma$ and $\theta$, respectively. Extending the conventional power notation, we let $\boldsymbol{\lambda^L} = \text{pow}(\boldsymbol{\lambda}, \boldsymbol{L})$ and $\boldsymbol{\gamma^L} = \text{pow}(\boldsymbol{\gamma}, \boldsymbol{L})$, i.e., $(\boldsymbol{\lambda^L})_{i,j} = \boldsymbol{\lambda}_{i,j}^{\boldsymbol{L}_{i,j}}$ and $(\boldsymbol{\gamma^L})_{i,j} = \boldsymbol{\gamma}_{i,j}^{\boldsymbol{L}_{i,j}}$. Then we can calculate the REMs by

$$\boldsymbol{P} = \begin{cases} \boldsymbol{D}^{\text{R}} - \boldsymbol{I}_T & \text{for regular REM,} \\ \boldsymbol{D}^{\text{C}i} - \boldsymbol{I}_T & \text{for cyclical REMs with } 1 \le i \le 2, \end{cases}$$

where $\boldsymbol{D}^{\text{R}} = \boldsymbol{\lambda^L}$, $\boldsymbol{D}^{\text{C1}} = \boldsymbol{\gamma^L} \odot \cos(\boldsymbol{\theta} \odot \boldsymbol{L})$, $\boldsymbol{D}^{\text{C2}} = \boldsymbol{\gamma^L} \odot \sin(\boldsymbol{\theta} \odot \boldsymbol{L})$ and $\odot$ is the Hadamard product. Such design allows us to circumvent redundant computations and achieve parallelization as much as possible. To avoid extremely large or small values, we mask the entries that have power larger than a preset number $K$. In practice, we set $K = 200$.

Moreover, we can easily extend the above calculation to dilated REMs by substituting $\boldsymbol{L}$ with $\boldsymbol{L}_d$, where $(\boldsymbol{L}_d)_{i,j} = \mathbb{I}_{\{\boldsymbol{L}_{i,j} \bmod d = 0\}} \boldsymbol{L}_{i,j}/d$ and $d$ is the dilating factor. The remaining operations are unchanged.

### D.2  FLEXIBLE ADAPTATION TO DIFFERENT ATTENTION SHAPES

In some cases, the query and key have different time spans with an overlapping period; examples including the cross-attention and all attention in the Transformer-XL. We show in this section that our REMs can be adapted for such cases. Typically, the historical information will be mostly stored in the query matrix, and the new information will be included in the key and the value matrices. Assume that the query matrix contains the tokens from time $1$ to time $T$, and the key matrix contains that from time $T - K$ to time $T + L$, where $K > 0$ is the overlapping period length and $L > 0$ is the length of new information. Then the REM is of size $(L + K + 1) \times T$ and takes the form of

$$\begin{array}{c} \\ \begin{array}{r} T-K \\ T-K+1 \\ \vdots \\ T-1 \\ T \\ T+1 \\ \vdots \\ T+L \end{array} \begin{pmatrix} \overset{1}{f_{T-K-1}(\beta)} & \overset{2}{f_{T-K-2}(\beta)} & \overset{\cdots}{\cdots} & \overset{T-K-1}{f_1(\beta)} & \overset{T-K}{0} & \overset{\cdots}{\cdots} & \overset{T-1}{f_{K-1}(\beta)} & \overset{T}{f_K(\beta)} \\ f_{T-K-2}(\beta) & f_{T-K-3}(\beta) & \cdots & f_2(\beta) & f_1(\beta) & \cdots & f_{K-2}(\beta) & f_{K-1}(\beta) \\ \vdots & \vdots & \ddots & \vdots & \vdots & \ddots & \vdots & \vdots \\ f_{T-2}(\beta) & f_{T-3}(\beta) & \cdots & f_K(\beta) & f_{K-1}(\beta) & \cdots & 0 & f_1(\beta) \\ f_{T-1}(\beta) & f_{T-2}(\beta) & \cdots & f_{K+1}(\beta) & f_K(\beta) & \cdots & f_1(\beta) & 0 \\ f_T(\beta) & f_{T-1}(\beta) & \cdots & f_{K+2}(\beta) & f_{K+1}(\beta) & \cdots & f_2(\beta) & f_1(\beta) \\ \vdots & \vdots & \ddots & \vdots & \vdots & \ddots & \vdots & \vdots \\ f_{T+L-1}(\beta) & f_{T+L-2}(\beta) & \cdots & f_{K+L+1}(\beta) & f_{K+L}(\beta) & \cdots & f_{L+1}(\beta) & f_L(\beta) \end{pmatrix}, \end{array}$$

which maintains the recurrent patterns while considering the shift in the relative time locations, where $\beta = \lambda$ for regular REM or $(\gamma, \theta)$ for cyclical REMs. And we can extend to the dilated REMs similarly.

### D.3  THE COMPUTATIONAL EFFICIENCY OF THE REMS FOR VARIOUS TRANSFORMER BASELINES

We first conduct an empirical study on the Weather dataset to evaluate the computational time of the REMs for RSA-Informer, RSA-LogSparse and RSA-XL. Specifically, we record the training time per

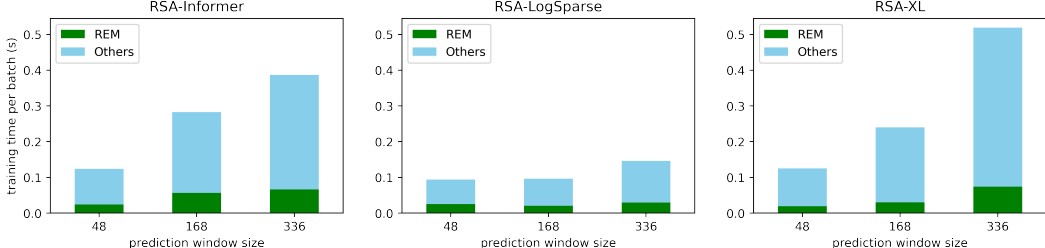

Figure 8: The training time ber batch for each RSA model, which can be decomposed into the additional computation time due to the REMs (green bar) and the others (blue bar).

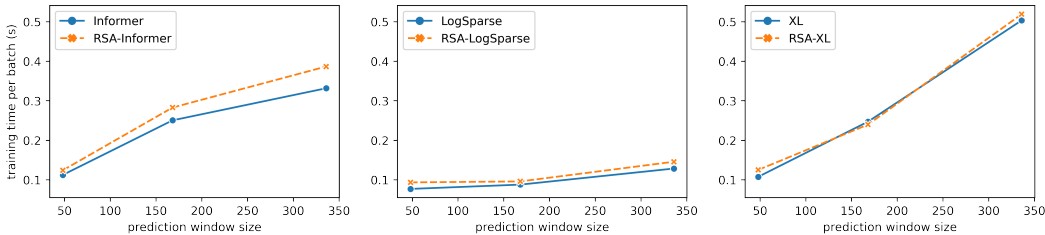

Figure 9: Comparison of the training time of ber batch between RSA-Informer and Informer (left panel), RSA-Logsparse Transformer and Logsparse Transformer (middle panel), RSA-Transformer-XL and Transformer-XL (right panel). The experiment is performed on Weather dataset.

batch for the RSA models, while simultaneously we exclude the REMs from the models and record the training time again. The difference between the two gives the additional computation time due to the REMs, as shown by the green bar in Figure 8. It can be seen that the calculations involving the REMs account for less than 20% of the total training time in general.

In the next experiment, we further compare the computational time between the RSA models and their respective baseline models. Note that the original Informer and LogSparse Transformer use absolute positional encoding (APE), while Transformer-XL adopts a learned relative positional encoding (RPE). These positional encoding schemes are excluded from the RSA models. The training time per batch for all the models are reported in Figure 9. We can observe a small increase in computational time for RSA-Informer and RSA-LogSparse against their baselines, while that for RSA-XL remains nearly unchanged. This implies that the REMs requires a bit more computation effort than the APEs, and the same computation effort as the RPE introduced by Transformer-XL.

## E ADDITIONAL EXPERIMENTS AND RESULTS

### E.1 TIME SERIES FORECASTING

We apply the same time series datasets as in the Informer paper (Zhou et al., 2021), and the details are given below for completeness. The first dataset is the Electricity Transformer Temperature (ETT). It contains the records for seven electric power indicators over a two-year period from two separated counties in China. Zhou et al. (2021) created three separate datasets $ETTh_1$ and $ETTh_2$ for hourly level, and $ETTm_1$ for the fifteen-minute level. The train/val/test is 12/4/4 months. The second dataset consists of climatological data over a four-year period from 2010 to 2013 for various locations in US. The train/val/test is 28/10/10 months. To ensure the quality of our baselines models, we make reference to the following two github repositories, namely *Informer2020*[4] and *transformer-xl*[5]. Since Transformer-XL requires a sequential order for training, testing and validation sets, we generate the training samples by sliding window starting from different points of the original time series following the practice in Salinas et al. (2020).

In this task, we let the REMs substitute the original positional encoding scheme in the RSA models. Moreover, since the distilling operation in the original Informer design is likely to distort the relative

---

[4]The link to the repository is https://github.com/zhouhaoyi/Informer2020 under Apache-2.0 license.

[5]The link to the repository is https://github.com/kimiyoung/transformer-xl under Apache-2.0 license.

Table 4: Multivariate long sequence time-series forecasting results on ETT and Weather datasets. The standard deviation of the 5 repetitions is included in brackets below the mean value. The results are reported on different prediction window sizes. We compare the MSE and MAE of RSA models with their baselines under all settings (a total of 32 comparisons), and report the number of times each model surpasses its comparing model in the last row.

| Methods | | Informer | | RPSA-Informer | | LogSparse | | RPSA-LogSparse | | XL | | RPSA-XL | |
|---|---|---|---|---|---|---|---|---|---|---|---|---|---|---|
| Metric | | MSE | MAE | MSE | MAE | MSE | MAE | MSE | MAE | MSE | MAE | MSE | MAE |
| ETTh1 | 24 | 0.762 | 0.632 | **0.414** | **0.450** | 1.124 | 0.826 | **0.858** | **0.668** | 0.514 | 0.518 | **0.466** | **0.493** |
| | | ( 0.144 ) | ( 0.076 ) | ( 0.014 ) | ( 0.012 ) | ( 0.012 ) | ( 0.014 ) | ( 0.052 ) | ( 0.034 ) | ( 0.031 ) | ( 0.018 ) | ( 0.003 ) | ( 0.002 ) |
| | 48 | 1.006 | 0.763 | **0.467** | **0.493** | 1.161 | 0.841 | **0.818** | **0.659** | 0.571 | 0.556 | **0.528** | **0.535** |
| | | ( 0.024 ) | ( 0.020 ) | ( 0.018 ) | ( 0.017 ) | ( 0.094 ) | ( 0.059 ) | ( 0.021 ) | ( 0.010 ) | ( 0.010 ) | ( 0.009 ) | ( 0.009 ) | ( 0.008 ) |
| | 168 | 1.141 | 0.823 | **0.753** | **0.659** | 1.104 | 0.818 | **1.042** | **0.781** | 0.898 | 0.734 | **0.813** | **0.694** |
| | | ( 0.022 ) | ( 0.023 ) | ( 0.036 ) | ( 0.018 ) | ( 0.010 ) | ( 0.016 ) | ( 0.021 ) | ( 0.007 ) | ( 0.010 ) | ( 0.006 ) | ( 0.011 ) | ( 0.008 ) |
| | 336 | 1.416 | 0.987 | **0.895** | **0.755** | 1.178 | 0.851 | **0.972** | **0.774** | 0.963 | 0.758 | **0.942** | **0.756** |
| | | ( 0.071 ) | ( 0.034 ) | ( 0.040 ) | ( 0.016 ) | ( 0.055 ) | ( 0.036 ) | ( 0.013 ) | ( 0.008 ) | ( 0.033 ) | ( 0.012 ) | ( 0.012 ) | ( 0.008 ) |
| ETTh2 | 24 | 2.558 | 1.253 | **1.264** | **0.879** | 2.894 | 1.375 | **1.076** | **0.844** | 0.763 | 0.699 | **0.705** | **0.660** |
| | | ( 1.177 ) | ( 0.275 ) | ( 0.479 ) | ( 0.135 ) | ( 0.131 ) | ( 0.029 ) | ( 0.034 ) | ( 0.017 ) | ( 0.042 ) | ( 0.021 ) | ( 0.050 ) | ( 0.018 ) |
| | 48 | 2.487 | 1.268 | **1.878** | **1.067** | 3.009 | 1.363 | **1.362** | **0.975** | 1.293 | 0.917 | **1.171** | **0.877** |
| | | ( 0.088 ) | ( 0.018 ) | ( 0.455 ) | ( 0.128 ) | ( 0.074 ) | ( 0.012 ) | ( 0.089 ) | ( 0.043 ) | ( 0.061 ) | ( 0.031 ) | ( 0.082 ) | ( 0.030 ) |
| | 168 | 2.869 | 1.324 | **2.830** | **1.301** | 2.876 | 1.307 | **2.165** | **1.235** | 2.780 | 1.288 | **2.671** | 1.290 |
| | | ( 0.124 ) | ( 0.019 ) | ( 0.188 ) | ( 0.039 ) | ( 0.068 ) | ( 0.012 ) | ( 0.131 ) | ( 0.030 ) | ( 0.157 ) | ( 0.045 ) | ( 0.133 ) | ( 0.026 ) |
| | 336 | **2.055** | **1.113** | 2.113 | 1.124 | 3.005 | **1.350** | **2.909** | 1.365 | **2.447** | **1.231** | 2.461 | 1.238 |
| | | ( 0.058 ) | ( 0.017 ) | ( 0.120 ) | ( 0.039 ) | ( 0.073 ) | ( 0.045 ) | ( 0.094 ) | ( 0.047 ) | ( 0.080 ) | ( 0.020 ) | ( 0.083 ) | ( 0.028 ) |
| ETTm1 | 24 | 0.536 | 0.511 | **0.534** | **0.507** | 1.105 | 0.837 | **0.619** | **0.553** | **0.561** | **0.537** | 0.591 | 0.549 |
| | | ( 0.023 ) | ( 0.014 ) | ( 0.023 ) | ( 0.011 ) | ( 0.053 ) | ( 0.034 ) | ( 0.063 ) | ( 0.031 ) | ( 0.021 ) | ( 0.011 ) | ( 0.012 ) | ( 0.011 ) |
| | 48 | 0.781 | 0.633 | **0.644** | **0.612** | 1.150 | 0.852 | **0.541** | **0.519** | 0.562 | 0.543 | **0.556** | **0.536** |
| | | ( 0.265 ) | ( 0.135 ) | ( 0.079 ) | ( 0.049 ) | ( 0.015 ) | ( 0.016 ) | ( 0.010 ) | ( 0.005 ) | ( 0.009 ) | ( 0.005 ) | ( 0.020 ) | ( 0.012 ) |
| | 96 | 0.823 | 0.697 | **0.732** | **0.665** | 1.227 | 0.897 | **0.526** | **0.520** | 0.714 | **0.640** | **0.707** | 0.645 |
| | | ( 0.095 ) | ( 0.052 ) | ( 0.079 ) | ( 0.051 ) | ( 0.057 ) | ( 0.035 ) | ( 0.090 ) | ( 0.057 ) | ( 0.075 ) | ( 0.049 ) | ( 0.045 ) | ( 0.029 ) |
| | 288 | 1.371 | 0.945 | **0.835** | **0.710** | 1.167 | 0.862 | **0.955** | **0.766** | 0.969 | 0.795 | **0.967** | **0.795** |
| | | ( 0.246 ) | ( 0.059 ) | ( 0.058 ) | ( 0.036 ) | ( 0.026 ) | ( 0.019 ) | ( 0.149 ) | ( 0.072 ) | ( 0.093 ) | ( 0.051 ) | ( 0.007 ) | ( 0.003 ) |
| Weather | 24 | **0.316** | **0.371** | 0.328 | 0.380 | 0.560 | 0.553 | **0.394** | **0.441** | **0.364** | **0.411** | 0.367 | 0.414 |
| | | ( 0.003 ) | ( 0.002 ) | ( 0.001 ) | ( 0.001 ) | ( 0.069 ) | ( 0.046 ) | ( 0.015 ) | ( 0.012 ) | ( 0.001 ) | ( 0.001 ) | ( 0.000 ) | ( 0.000 ) |
| | 48 | 0.606 | 0.566 | **0.432** | **0.464** | 0.582 | 0.567 | **0.432** | **0.466** | 0.473 | 0.490 | **0.466** | **0.484** |
| | | ( 0.194 ) | ( 0.100 ) | ( 0.004 ) | ( 0.002 ) | ( 0.029 ) | ( 0.021 ) | ( 0.012 ) | ( 0.009 ) | ( 0.002 ) | ( 0.002 ) | ( 0.001 ) | ( 0.001 ) |
| | 168 | 1.009 | 0.771 | **0.862** | **0.702** | 0.929 | 0.754 | **0.602** | **0.580** | 0.684 | 0.616 | **0.592** | **0.572** |
| | | ( 0.006 ) | ( 0.003 ) | ( 0.189 ) | ( 0.088 ) | ( 0.200 ) | ( 0.101 ) | ( 0.027 ) | ( 0.017 ) | ( 0.176 ) | ( 0.085 ) | ( 0.017 ) | ( 0.012 ) |
| | 336 | 1.096 | 0.801 | **0.846** | **0.697** | 0.874 | 0.734 | **0.638** | **0.602** | 0.895 | 0.713 | **0.816** | **0.679** |
| | | ( 0.236 ) | ( 0.069 ) | ( 0.187 ) | ( 0.084 ) | ( 0.132 ) | ( 0.071 ) | ( 0.025 ) | ( 0.014 ) | ( 0.155 ) | ( 0.071 ) | ( 0.167 ) | ( 0.079 ) |
| count | | 4 | | 28 | | 1 | | 31 | | 8 | | 24 | |

positions, we exclude it from both our Informer baseline and the RSA-Informer. The hyperparameters in RSA are fine tuned via grid search.

Table 5 reports the fine-tuned hyperparameter settings considered in Table 1 of the main paper. To provide additional information about the standard deviation, we provide a more detailed version of Table 1 of the main paper, given in Table 4.

**Ablation on the hyperparameters** To shed light on the importance of different recurrent dynamics, we provide an ablation study with RSA-Informer on ETTh$_1$. The model has a total of 8 heads, and we set $k_2 = k_3$ to some positive integer $s$ and $k_5 = k_6$ to some positive integer $\bar{s}$. In particular, when $\bar{s} = 0$, we also take $k_4 = 0$ and hence there are not any dilated REMs, leading to $k_1 = 8 - 2s$. Otherwise, we take $k_4 = 4 - 2\bar{s}$ and $k_1 = 4 - 2s$. It can be observed from Table 6 that the inclusion of dilation and the cyclical patterns can significantly improve the forecasting performance. The concurrence of dilated and non-dilated RSA heads allows the model to capture both the short- and long-term recurrent dynamics.

**Gate-control parameter learning** To study how much proportion of the RSA output is accounted for by the REMs, Figure 10 plots the change in its relative importance i.e. $\sigma(\mu)$ during training for all the attention layers, starting from different initializations. As the learning rate drops rapidly, all curves are flattened towards the end. The relative positional attention is the most effective for the cross-attention layer, and the least effective for the self-attention layer in the decoder, whose recurrent patterns may be disrupted by zero paddings. We observe a decrease in MSE as the initialization increases from $-1$ to $1$, which provides an empirical support for initializing at $\mu = 1$.

Table 5: The fine-tuned hyperparameter settings used for the RSA models in Table 1. The tuple $(k_1, k_2, k_3, k_4, k_5, k_6)$ refer to the number of the six types of REMs. The values of different dilating factors are listed, where those for regular heads are followed by those for the cyclical heads. $d =$ "$-$" means no dilated REMs are considered.

| Methods | | RSA-Informer | | RSA-LogSparse | | RSA-XL | |
|---|---|---|---|---|---|---|---|
| Setting | | $(k_1,k_2,k_3,k_4,k_5,k_6)$ | $d$ | $(k_1,k_2,k_3,k_4,k_5,k_6)$ | $d$ | $(k_1,k_2,k_3,k_4,k_5,k_6)$ | $d$ |
| ETTh1 | 24 | (0,2,2,0,2,2) | [24,24,24,24] | (4,0,0,4,0,0) | [24,24,24,24] | (4,0,0,4,0,0) | [24,24,24,24] |
| | 48 | (0,2,2,2,1,1) | [24,24,24,24] | (4,0,0,4,0,0) | [24,24,24,24] | (4,0,0,4,0,0) | [24,24,24,24] |
| | 168 | (0,2,2,2,1,1) | [24,24,24,24] | (4,0,0,4,0,0) | [24,24,24,24] | (0,2,2,0,2,2) | [48,24,48,24] |
| | 336 | (4,0,0,4,0,0) | [24,24,24,24] | (0,2,2,0,2,2) | [48,24,48,24] | (1,2,2,1,1,1) | [48,24,24] |
| ETTh2 | 24 | (4,2,2,0,0,0) | - | (4,0,0,4,0,0) | [24,24,24,24] | (4,0,0,4,0,0) | [24,24,24,24] |
| | 48 | (0,2,2,0,2,2) | [24,24,24,24] | (8,0,0,0,0,0) | - | (0,2,2,0,2,2) | [48,24,48,24] |
| | 168 | (4,2,2,0,0,0) | - | (4,0,0,4,0,0) | [24,24,24,24] | (4,0,0,4,0,0) | [24,24,24,24] |
| | 336 | (0,2,2,0,2,2) | [24,24,24,24] | (4,0,0,4,0,0) | [24,24,24,24] | (4,0,0,4,0,0) | [24,24,24,24] |
| ETTm1 | 24 | (0,0,0,0,4,4) | [2,3,6,24,2,3,6,24] | (4,0,0,4,0,0) | [96,96,96,96] | (0,1,1,0,3,3) | [48,24,24,48,24,24] |
| | 48 | (0,2,2,0,2,2) | [96,96,96,96] | (4,0,0,4,0,0) | [96,96,96,96] | (0,1,1,2,2,2) | [48,24,48,96,48,96] |
| | 96 | (0,2,2,0,2,2) | [96,96,96,96] | (4,0,0,4,0,0) | [96,96,96,96] | (0,1,1,2,2,2) | [48,24,48,96,48,96] |
| | 288 | (0,2,2,0,2,2) | [96,96,96,96] | (4,0,0,4,0,0) | [96,96,96,96] | (0,1,1,2,2,2) | [48,24,48,96,48,96] |
| Weather | 24 | (0,0,0,0,4,4) | [3,5,7,24,3,5,7,24] | (4,0,0,4,0,0) | [24,24,24,24] | (0,2,2,0,2,2) | [24,24,24,24] |
| | 48 | (0,2,2,0,2,2) | [24,24,24,24] | (4,0,0,4,0,0) | [24,24,24,24] | (0,2,2,0,2,2) | [48,24,48,24] |
| | 168 | (0,2,2,0,2,2) | [24,24,24,24] | (4,0,0,4,0,0) | [24,24,24,24] | (1,2,2,1,1,1) | [48,24,24] |
| | 336 | (0,2,2,0,2,2) | [24,24,24,24] | (8,0,0,0,0,0) | - | (0,2,2,0,2,2) | [48,24,48,24] |

Table 6: Ablation study on the hyperparameters for the 8-headed RSA-Informer on ETTh$_1$ dataset. $\bar{s}$ and $s$ correspond to the number of dilated and non-dilated cyclical REMs. And $\bar{s} =$"-" refers to the case where no dilated REMs are considered. Otherwise, 4 out of 8 dilated REMs are considered. The MSEs are averaged over 5 repetitions.

| $\bar{s}$ \ $s$ | 0 | 1 | 2 | | $\bar{s}$ \ $s$ | 0 | 1 | 2 | | $\bar{s}$ \ $s$ | 0 | 1 | 2 | | $\bar{s}$ \ $s$ | 0 | 1 | 2 |
|---|---|---|---|---|---|---|---|---|---|---|---|---|---|---|---|---|---|---|
| - | 0.997 | 0.962 | 0.878 | | - | 1.071 | 1.072 | 1.066 | | - | 1.254 | 1.183 | 1.089 | | - | 1.507 | 1.382 | 1.239 |
| 0 | 0.455 | 0.472 | 0.430 | | 0 | 0.586 | 0.561 | 0.572 | | 0 | 0.800 | 0.795 | 0.994 | | 0 | **0.895** | 1.048 | 0.946 |
| 1 | 0.475 | 0.505 | **0.414** | | 1 | 0.558 | 0.548 | 0.584 | | 1 | 0.772 | 0.806 | 1.005 | | 1 | 1.076 | 1.150 | **0.895** |
| 2 | 0.521 | 0.511 | 0.422 | | 2 | 0.473 | **0.467** | 0.469 | | 2 | 0.777 | **0.753** | 0.879 | | 2 | 1.192 | 1.198 | 1.060 |
| pred window = 24 | | | | | pred window = 48 | | | | | pred window = 168 | | | | | pred window = 336 | | | |

## E.2 REGULAR LANGUAGE LEARNING

The six types of regular language datasets are obtained from the github repository *Transformer-Formal-Language*.[6] We briefly describe the rules of these languages:

- The Parity language contains the numbers $\{0, 1\}$ with an even number of 1's. For example, the string 0110 belongs to the language but 0111 does not.
- The $D_n$ language is defined on the alphabets $\{a,b\}$. The $D_1$ language can be written using the regular expression $(ab)^\star$, where "$\star$" represents the Kleene star operation. Then for $n \geq 1$, $D_n$ can be expressed by the recursive relationship $D_n = (aD_{n-1}b)^\star$. We choose $D_2$ and $D_4$, where $D_2 = (a(ab)^\star b)$ and $D_4 = (a(a(a(ab)^\star b)^\star b)^\star b)^\star$.
- The Tomita languages also use the numbers $\{0, 1\}$ and include seven different grammars. In our experiment, we choose Tomita 3, 5 and 6. In Tomita 3, consecutive 1's must be of an odd number and consecutive 0's must be of an even number. In Tomita 5, the total numbers of 1's and 0's must both be even. Finally, Tomita 6 requires the difference between the total number of 1's and 0's to be a multiple of 3.

In this experiment, we adopt a simple setting with 3 layers and 5 heads for all models. The embedding size is set to 20. During training, the Adam optimizer is initialized at 0.005 with a $0.5\times$ decay every 5 epochs. And all models are run for 25 epochs without early stopping. Both the training set and bin 0 contain samples with length in the range of $[2, 50]$ for Parity and Tomita, and $[2, 100]$ for $D_n$, while bin 1 contains longer test samples with length in the range of $[51, 100]$ for Parity and Tomita, and $[101, 200]$ for $D_n$. The models are trained on 10K samples and tested on 2K samples per bin for Parity and Tomita, and trained on 5K samples with 1K test samples per bin for $D_n$. Following Bhattamishra et al. (2020), we train the models as below: starting from the first position of an input sequence, the model outputs a string of possible characters for the next position, and this continues

---

[6]The link to the repository is https://github.com/satwik77/Transformer-Formal-Languages under MIT license.

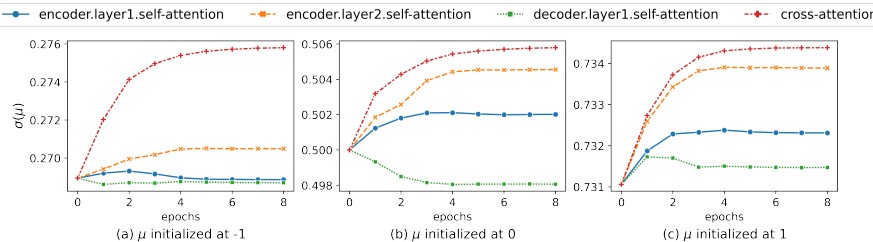

Figure 10: (a) − (c) report the change in $\sigma(\mu)$ at each attention layer during training, where $\mu$ is initialized at $-1, 0$, and $1$, respectively. An 8-headed RSA-Informer is applied to the ETTh$_1$ dataset with settings reported in Table 5. The prediction window size equals to $24$.

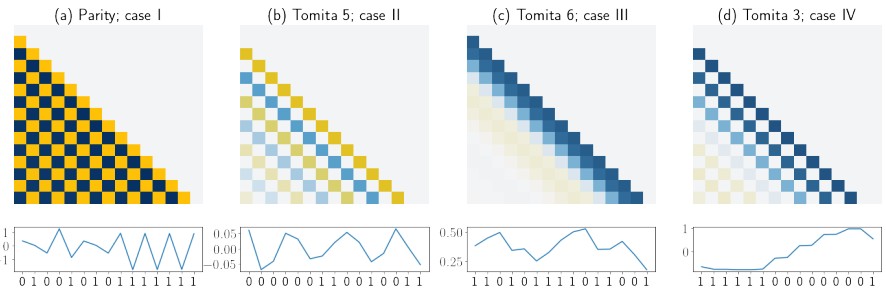

Figure 11: We visualize four recurrent patterns provided by the RSA heads on different languages (top), and their corresponding output, averaged over embedding dimension (bottom). The corresponding input sequence is given on $x$-axis.

until the sequence is exhausted. The prediction task is recast into a classfication task, where the prediction is deemed a success if and only if the output values at each position are correct, and a failure otherwise.

From Table 2, we can see that the Transformer-only models may require much more training samples to learn these patterns. Moreover, when the training samples are shorter than the test samples, they will have difficulty in extending the learned patterns to longer sequences, even with absolute or learned relative positional encoding. Meanwhile, REMs can easily extend its learned patterns to any sequence length, which may account for the better generalization ability of RSA-Transformers. To help interpret the results in Table 2, we provide some visualizations of the learned recurrent patterns on different languages in Figure 11. Learning Parity requires the model to count the number of 1's, and then decide whether it is an even number. Its recurrent pattern at (a) mainly consists of alternating $\{-1, 1\}$ that leads to regular oscillations when encountering an even or odd number of 1's. Tomita 5 has an additional requirement that the number of 0 is also even, and hence its learning can be assisted by dilated patterns; see (b) for illustration. Although Tomita 6 is more complicated, the regular cyclical patterns at (c) seems sufficient for it to learn well. This gives us a glimpse of how the RSA-Transformer manages to learn the regular languages.

### E.3 CODE AND NATURAL LANGUAGE MODELING

For RSA-CodeT5-small, our code is based on the original CodeT5, which can be referred to in the repository `https://github.com/salesforce/CodeT5`.

For the RSA-XL models adopted for the natural language modeling, only some of the heads are replaced with the RSA heads, while the remaining ones are still the original XL heads.

In the RSA-BRT and baseline BRT models, the recurrent layer is put at the third layer counted from the top, i.e., the recurrent layer is at layer 12 for 14-layer models and at layer 14 for 16-layer models. The nonrecurrent layers are of the same structure as RSA-XL and XL respectively.

In Table 7, we report the detailed hyperparameters settings for the RSA and their baseline models used in Table 3, where "n_RSA_heads" refers to the number of heads that are changed to the RSA heads.

Table 7: The hyperparameter settings for the RSA and their baseline models used in Table 3.

| Dataset | Model | n_layer | n_heads | n_RSA_heads | $(k_1, k_2, k_3, k_4, k_5, k_6)$ | $d$ | $\mu$_init | #Params |
|---|---|---|---|---|---|---|---|---|
| Devign | CodeT5-small | 12 | 8 | - | - | - | - | 60492288 |
| | RSA-CodeT5-small | 12 | 8 | 8 | (0,0,0,8,0,0) | 64 | -3 | 60492396 |
| Text8 | XL | 14 | 8 | - | - | - | - | 47789595 |
| | RSA-XL | 14 | 8 | 4 | (0,0,0,0,2,2) | 6,6,12,12 | 1 | 47789665 |
| | BRT | 14 | 8 | - | - | - | - | 55464987 |
| | RSA-BRT | 14 | 8 | 4 | (0,0,0,0,2,2) | 5,5,10,10 | -1 | 55465052 |
| Enwik8 | XL | 14 | 8 | - | - | - | - | 47880396 |
| | RSA-XL | 14 | 8 | 4 | (0,0,0,0,2,2) | 3,6,9,12 | 1 | 47880466 |
| | BRT | 14 | 8 | - | - | - | - | 55555788 |
| | RSA-BR | 14 | 8 | 4 | (0,0,0,0,2,2) | 3,6,9,12 | 0 | 55555853 |
| WikiText-103 | XL | 16 | 12 | - | - | - | - | 153797302 |
| | RSA-XL | 16 | 12 | 6 | (0,0,0,2,2,2) | 12,24,12,24,12,24 | -1 | 153797414 |
| | BRT | 16 | 12 | - | - | - | - | 159627039 |
| | RSA-BR | 16 | 12 | 6 | (0,0,0,2,2,2) | 12,24,12,24,12,24 | -2 | 159627144 |

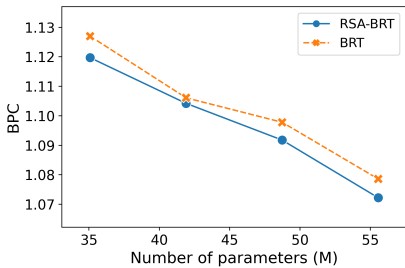

Figure 12: Bit-per-character for BRT and RSA-BRT with respect to the model size (i.e. number of parameters in millions) on Enwik8.

### E.4 ABLATION STUDY

This subsection contains one additional ablation experiment to compare the parameter efficiency of Block Recurrent Transformer (BRT) against its RSA variant i.e. RSA-BRT, and implementation details of the sequence length ablation study.

We conduct an additional experiment to compare the scaling performance of RSA-BRT and BRT with respect to the number of parameters on Enwik8 dataset. Specifically, the number of layers in both models varies in [8, 10, 12, 14], and the training schemes for NLP task in Section 4 is adopted. From the results presented in Figure 12, it can be observed that RSA-BRT consistently achieves better performance than its baseline.

Next, we provide implementation details for sequence length scaling experiement. For all four models (i.e. BRT, TLB, RSA-BRT, RSA-TLB), a 4 layer transformer with 8 attention heads are trained using a batch size of 16 and the Adam optimizer with a learning rate of 0.0001 which is reduced by half every 5 epochs for BRT and RSA-BRT.

