# OpenReview forum: "Encoding Recurrence into Transformers"
_ICLR.cc/2023/Conference — ICLR 2023 notable top 5%_

### Official Review · Reviewer_ZrmK · 2022-10-19

**Confidence:** 3
**Correctness:** 3
**Technical Novelty And Significance:** 3
**Empirical Novelty And Significance:** 3
**Recommendation:** 8

**Clarity, Quality, Novelty And Reproducibility:**

The presentation is easy to follow. I've read the theoretical justification and found no major issues. The method is somewhat novel in that it shows the equivalence between (linear) RNN and self-attention, although similar results have been demonstrated in other areas such as vision, for example, the equivalence between convolution and self-attention.

**Strength And Weaknesses:**

+ The motivation is clear and the algorithm is sensible.

+ The proposed method is tested on several benchmarks.

- RNN block discussed in the paper is basically a masked linear aggregation of the tokens (``masked'' means each token can only attend to the previous tokens), with the aggregation weights ($P_{mask}$) specially designed. It would be helpful if the authors can compare to the baseline where the $P_{mask}$ is learnable, i.e., using a learnable masked linear aggregation. Another baseline would be a learnable unmasked linear aggregation. These comparisons can tell us if it's the RNN formulation that matters or just the linear aggregation.

- The authors argue that, transformers on small-scale recurrent data will overfit, thus introducing the RNN module which has a better inductive bias can help prevent the overfitting. However, there is a learnable gate-control parameter to decide whether the model should rely more on self-attention or RNN. Won't that encourage the model to rely more on self-attention in order to overfit the training data?

- Since the authors use a linear RNN, I wonder how much model capacity are we losing by linearizing the RNN, i.e., how huge is the gap between linear and non-linear RNN, performance-wise?

- In Section 3, the authors denote by $p_{total}$ the total temporal patterns in the data. How is this value defined?

- Section 2.1, $b \in \mathbb{R}^{d_{in}}$ -> $b \in \mathbb{R}^{d}.



**Summary Of The Paper:**

For small-scaled sequential data, transformers tend to overfit, while RNNs has better inductive bias that prevents the overfitting. However, RNN can't be trained in a parallel way like transformers. In this work, the authors find that a linear RNN has a simple form of masked linear aggregation, which can be formulated as a specific self-attention module, thus can be easily incorporated in to transformers and jointly trained in a parallel way. The authors propose to combine regular self-attention and RNN module together, and use a gating function to decide the weight of the two paths, making it possible to fit data with different scales. Experiments on several sequential modeling tasks show the advantage of the proposed combination of transformers and RNNs.

**Summary Of The Review:**

Overall the paper is solid in both theoretical and empirical parts, although comparisons to the baseline models are missing.

---

> ### Author Response · Authors · 2022-11-17
> **Thanks for your comments!**
>
> Thanks for your constructive comments and suggestions, and they are exceedingly helpful to improve our paper. We have carefully incorporated them in the revised paper. In the following, your comments are first stated and then followed by our point-by-point responses.
>
> > 1. RNN block discussed in the paper is basically a masked linear aggregation of the tokens ("masked'' means each token can only attend to the previous tokens), with the aggregation weights $P_{mask}$ specially designed. It would be helpful if the authors can compare to the baseline where the $P_{mask}$ is learnable, i.e., using a learnable masked linear aggregation. Another baseline would be a learnable unmasked linear aggregation. These comparisons can tell us if it's the RNN formulation that matters or just the linear aggregation.
>
> Thank you very much for this insightful comment. Yes, comparing with a fully learnable aggregation weight (denoted by $P_{\text{FL}}$), the proposed REM (denoted by $P_{\text{RSA}}$ to distinguish from the other types of linear aggregation) can be treated as a special parametric form with one or two additional parameters only. In fact, the REM is also a special case of a learnable Toeplitz matrix (denoted by $P_{\text{TL}}$). Note that a fully learnable or Toeplitz matrix has already included all kinds of temporal decay patterns automatically, and hence an advantage of using $P_{\text{FL}}$ and $P_{\text{TL}}$ is that we do not need to specify the numbers of different temporal decay patterns.
>
> In spite of their generality, the fully learnable $P_{\text{FL}}$ and Toeplitz $P_{\text{TL}}$ matrices will require $O(T^2)$ and $O(T)$ additional parameters, respectively, where $T$ is the sequence length. More importantly, the proposed RSA also includes a standard self-attention, which is flexible enough for all remaining patterns from the fitted $P_{\text{RSA}}$. As a result, it may not be necessary to consider a more general yet less efficient structure for $P$. Moreover, RSA combines the recurrent features of the RNN, making it particularly suitable for sequential learning problems.
>
> Following your suggestions, we have conducted an ablation study in Section 4.4 (Page 8-9) on Text8 dataset to compare the proposed RSA module against a modified XL (FL-XL) (i.e. $P_{\text{RSA}}$ is replaced by $P_{\text{FL}}$). Since this is a prediction task, both $P_{\text{RSA}}$ and $P_{\text{FL}}$ are set to be masked. To obtain more insights, we have scaled the model sizes by varying the number of layers, and the performance is evaluated in terms of BPC. The detailed results are given below, and they have also been plotted in Figure 4 (left panel) of the paper.
>
>
> | #  layers |              | 8     | 10        | 12        | 14        | 16        |
> | --------- | ------------ | ----- | --------- | --------- | --------- | --------- |
> | XL        | #Params  (M) | /     | 34.14     | 40.96     | 47.79     | 54.61     |
> |           | BPC          | /     | 1.196     | 1.184     | 1.171     | 1.170     |
> | RSA-XL    | #Params  (M) | /     | 34.14     | 40.96    | 47.79     | 54.61     |
> |           | BPC          | /     | **1.181** | **1.170** | **1.164** | **1.160** |
> | FL-XL     | #Params  (M) | 35.72 | 44.65     | 53.58     | 62.51     | /         |
> |           | BPC          | 1.214 | 1.196     | 1.189     | 1.182     | /         |
>
> From the above table or Figure 4 in the paper, we have two findings below: (1) the inclusion of $P_{\text{FL}}$ may not be useful in improving the baseline performance, since the fully learnable matrix may lead to model redundancy, (2) RSA-XL is the most sample-efficient design among the three models, namely it achieves the best performance when the number of parameters is comparable to the other two models.

---

> > ### Author Response · Authors · 2022-11-17
> > **Additional reply to the first comment**
> >
> > As you suggested, we have also conducted experiments to compare the unmasked $P_{\text{RSA}}$, unmasked $P_{\text{FL}}$ and unmasked  $P_{TL}$ on the code defect detection task in Section 4.3, since it uses non-causal full attention. Same as in the paper, we adapted the pretrained CodeT5-small to the downstream code defect detection task, and modified its self-attention by $P_{\text{RSA}}, P_{\text{FL}}$ and $P_{\text{FL}}$, respectively. All models have been fine-tuned for 20 epochs with early stopping. The results are reported below for your reference.
> >
> > |                                                      | CodeT5-small | RSA-CodeT5-small | TL-CodeT5-small | FL-CodeT5-small |
> > | ---------------------------------------------------- | ------------ | ---------------- | --------------- | --------------- |
> > | Defect Accuracy (%)                                  | 64.6         | **65.96**        | 56.66           | 52.75           |
> > | Increase in #Params against CodeT5-small             | 0            | 108              | 24,588          | 3,145,734       |
> > | Percentage increase in #Params against  CodeT5-small | 0            | 0.0002%          | 0.0406%         | 5.2002%         |
> >
> > It can be seen that TL and FL introduce ~200 times and ~30000 times more new parameters than the baseline CodeT5-small. This may explain the poor performance of these two models when they are used jointly with pretrained models.
> >
> > Moreover, we are currently conducting experiments to compare the performance of RSA against TL on the Text8 dataset. These results will be provided soon, and we will add them to Section 4.4. In the meanwhile, we may expect that the RSA can still beat TL in a similar way.
> >
> > **Summary of paper revision related to this comment**
> >
> > (1) A discussion has been added to the paragraph "More discussions on REMs" in Section 3  on Page 6 to compare REM to other types of linear aggregation weight matrices.
> >
> > (2) An ablation study has been conducted to study the parameter-efficiency of RSA, and it has been added to Section 4.4 on Page 8-9 with results shown in Figure 4 (left panel).

---

> > > ### Author Response · Authors · 2022-11-17
> > > **Reply to the second comment**
> > >
> > > > 2. The authors argue that, transformers on small-scale recurrent data will overfit, thus introducing the RNN module which has a better inductive bias can help prevent the overfitting. However, there is a learnable gate-control parameter to decide whether the model should rely more on self attention or RNN. Won't that encourage the model to rely more on self-attention in order to overfit the training data?
> > >
> > > Thanks very much for this insightful comment, and it actually pushes us to think about the roles of $\sigma(\mu)$ more deeply.
> > >
> > > Firstly, the proposed RSA is trying to balance the RNN and self-attention by using the learnable gate $\sigma(\mu)$. As a result, when the sample size is small, the RNN will perform best due to the inductive bias, while the RSA performs slightly worse by including a small proportion of self-attention. However, for a relatively large sample size, the performance of RSA will be better than the RNN due to the flexibility of self-attention.
> > >
> > > Secondly, although the gate $\sigma(\mu)$ is learnable, it will automatically choose a larger proportion of RNNs during the training when the sample size is small. More specifically, most of Transformer models are trained by gradient-based methods, e.g. Adam or SGD, where the learnable parameters move towards the direction of the steepest loss descent. Since the gate value $\sigma(\mu)$ controls the weights assigned to self-attention and REM (i.e. RNN), a slight change in $\sigma(\mu)$ can change the loss by a large margin. As a result, $\sigma(\mu)$ is the key player to ensure that the model can perform well even under a small sample size, or in other words, it can achieve better sample efficiency. Theoretically speaking, since RNN enjoys better sample efficiency than Transformer (see illustration in Figure 1), $\sigma(\mu)$ will favor the RNN over the self-attention when the sample size is small, resulting in a higher weight assigned to the RNN.
> > >
> > > Thirdly, to verify the above arguments empirically, we have added an ablation study on the sample-efficiency of RSA against its baseline Transformer for Enwik8 dataset. In particular, we only use a restricted portion of the training data, and observe how $\sigma(\mu)$ changes as the training data shrink in size. The results are shown in Figure 4 (middle&right panels) of the paper, and we also present them below for your easy reference:
> > >
> > > | Proportion  of training data (%)              | 30        | 50        | 80        | 100       |
> > > | --------------------------------------------- | --------- | --------- | --------- | --------- |
> > > | XL                                            | 1.291     | 1.190     | 1.109     | 1.077     |
> > > | RSA-XL                                        | **1.280** | **1.187** | **1.101** | **1.068** |
> > > | maximum value of $\sigma(\mu)$ for all layers | 0.53      | 0.48      | 0.44      | 0.33      |
> > >
> > > It can be seen that, the maximum value of $\sigma(\mu)$ indeed increases when sample size decreases. In particular, when only 30% of the training samples are available, RNN has a weight of more than 50% in some layers. As a result, RSA-XL consistently achieves a better sample efficiency than the baseline XL model.
> > >
> > > **Summary of paper revision related to this comment**
> > >
> > > An ablation study on the sample-efficiency of RSA has been added in Section 4.4 on Page 8-9.

---

> > > > ### Author Response · Authors · 2022-11-17
> > > > **Reply to the third comment (Part I. theoretical gap)**
> > > >
> > > > > 3. Since the authors use a linear RNN, I wonder how much model capacity are we losing by linearizing the RNN, i.e., how huge is the gap between linear and non-linear RNN, performance-wise?
> > > >
> > > > Thanks for this very helpful comment. Following your suggestions, we have investigated the gaps both theoretically and empirically when using a linear RNN to approximate a nonlinear RNN, and a new section has been added to the Appendix; see Appendix B (Page 13-15).
> > > >
> > > > Shortly speaking, a non-ignorable gap will exist when a nonlinear activation is misspecified to be a linear one, as the REM does in the paper. However, the proposed RSA also includes a standard self-attention, which is flexible enough for all remaining signals, and it can remedy the problem to some extent; see Figure 6 in the Appendix.
> > > >
> > > > The main part of the paper has also been revised accordingly. Specifically, a short discussion has been included in the paragraph "More discussions on REMs" in Section 3  on Page 6.
> > > >
> > > > In addition, for your easy reference, we have summarized Appendix B below.
> > > >
> > > > (Theoretical Gap). In Section B.1, we have derived the prediction error when a linear RNN model is used to train the data generated by a nonlinear RNN. The mean squared prediction error is defined as
> > > > $$
> > > > e_{\text{pred}} := \min_{\mathbf{\theta}} \mathbb{E}(y_t - h_t(\mathbf{\theta}))^2,
> > > > $$
> > > > where $y_t = g_t + \varepsilon_t$ is the sequence generated from the nonlinear RNN,
> > > > $$
> > > > g_t = \sigma_h(w_h^* g_{t-1} + w_x^* x_t + b^*) \quad\text{with}\hspace{2mm}\sigma_h \hspace{2mm}\text{being the nonlinear activation function,}
> > > > $$
> > > > $\varepsilon_t$ is an additive error term with mean 0 and variance $\gamma$, and $h_t(\theta)$ is a linear RNN model with $\theta$ representing its unknown parameters.
> > > >
> > > > **Proposition 3. (simplified version)** If $|w^*_h|<1$ and some regularity conditions hold, then
> > > > $$
> > > > e\_{\text{pred}} \leq \underbrace{C(1-|w^*_h|)^{-1}}\_{\text{misspecification error}} + \underbrace{\gamma}\_{\text{irreducible system error}},
> > > > $$
> > > > where the first part is due to the misspecification of using the linear activation to approximate.
> > > >
> > > > Its detailed proof is given in Section B.1.

---

> > > > > ### Author Response · Authors · 2022-11-17
> > > > > **Reply to the third comment (Part II. empirical gap)**
> > > > >
> > > > > (Empirical Gap). In Section B.2, we have conducted a synthetic experiment to evaluate the performance of linear RNNs when there is nonlinearity in the data. We first generate the data by using a two-layer nonlinear RNN model where the first layer has the form of
> > > > > $$
> > > > > \mathbf{z}_t = \alpha \mathbf{g}_t + (1-\alpha) \mathbf{h}_t \hspace{3mm}\text{with}\hspace{3mm}
> > > > > 	\mathbf{h}_t = \mathbf{W}_h  \mathbf{z}\_{t-1} + \mathbf{W}_x  \mathbf{x}_t + \mathbf{b} \hspace{3mm}\text{and}\hspace{3mm}
> > > > > 	\mathbf{g}_t = \sigma_h(\mathbf{W}_h  \mathbf{z}\_{t-1} + \mathbf{W}_x  \mathbf{x}_t + \mathbf{b} ),
> > > > > $$
> > > > > and $\sigma_h(\cdot)$ is a nonlinear activation function, and $0\leq \alpha\leq 1$ is the weight of nonlinearity. The second layer is defined similarly with $\mathbf{x}_t$ being replaced by the output $\mathbf{z}_t$ of the first layer.
> > > > >
> > > > > As $\alpha$ increases from 0 to 1, the data generating process gradually changes from a strictly linear RNN to a nonlinear one, i.e. $\alpha$ essentially controls the proportion of nonlinearity involved. The generated sequence is fitted separately by a linear RNN, a nonlinear RNN with the corresponding activation, and a linear RNN combined with a self-attention, (i.e. the proposed RSA).
> > > > >
> > > > > The mean squared prediction error (MSPE) averaged over different repetitions are denoted by $e_{\text{pred}}^{\mathrm{L}}, e_{\text{pred}}^{\mathrm{NL}}$ and $e_{\text{pred}}^{\mathrm{RSA}}$ for the three models, respectively. Using nonlinear RNNs as the benchmark, the MSPE ratio for the linear RNN or the RSA is defined as
> > > > > $$
> > > > > \text{MSPE ratio for model }i = \frac{e_{ \text{pred}}^{i} }{ e_{ \text{pred} }^{ \text{NL} } },\quad\text{where }i \in \\{\text{L}, \text{RSA}\\}.
> > > > > $$
> > > > > Specifically speaking, if MSPE ratio is larger than one, then linear RNN or RSA performs worse than the nonlinear RNN. The obtained MPSE ratios under different activations have been plotted in Figure 6 on Page 15, and they are also presented in the table below.
> > > > >
> > > > > | $\alpha$ |  | 0         | 0.2       | 0.4       | 0.6       | 0.8       | 1         |
> > > > > | -------- | ------ | --------- | --------- | --------- | --------- | --------- | --------- |
> > > > > | Tanh     | Linear | **0.982** | 0.993     | 1.000     | 1.008     | 1.011     | 1.012     |
> > > > > |          | RSA    | 0.987     | **0.991** | **0.998** | **1.004** | **1.004** | **1.008** |
> > > > > | Sigmoid  | Linear | **0.882** | **0.967** | 1.004     | 1.020     | 1.022     | 1.037     |
> > > > > |          | RSA    | 0.892     | 0.976     | **1.001** | **1.000** | **1.007** | **1.015** |
> > > > > | Relu     | Linear | **0.941** | 0.968     | 0.988     | 1.012     | 1.014     | 1.014     |
> > > > > |          | RSA    | 0.951     | **0.952** | **0.987** | **1.002** | **1.004** | **1.002** |
> > > > >
> > > > > It can be seen that, when $\alpha=1$, nonlinear RNNs perform the best, while linear RNNs suffer from misspecification error. Alternatively, when $\alpha=0$, the opposite can be observed. Moreover, as $\alpha$ increases, i.e. there are more nonlinearity, it is expected that linear RNNs become less favorable, while the proposed RSA can remedy the problem to some extent. Especially when $\alpha>0.6$, the RSA consistently achieves better prediction performance than the pure linear RNN.
> > > > >
> > > > > **Summary of paper revision related to this comment**
> > > > >
> > > > > (1) A short discussion has been included in the paragraph "More discussions on REMs" in Section 3  on Page 6.
> > > > >
> > > > > (2) Appendix B (Page 13-15) has been added to provide a detailed depiction of the theoretical and empirical gaps between linear and nonlinear RNNs.

---

> > > > > > ### Author Response · Authors · 2022-11-17
> > > > > > **Reply to comments 4 and 5**
> > > > > >
> > > > > > > 4. In Section 3, the authors denote by the total temporal patterns in the data. How is this value defined?
> > > > > >
> > > > > > This notation has no explicit definition, and our original aim was to make explanations more clear. But it seems to cause confusions instead.
> > > > > > We are very sorry about it, and have removed these notations from the paper.
> > > > > >
> > > > > >
> > > > > >
> > > > > >
> > > > > > > 5. Section 2.1, $b \in \mathbb{R}^{d_{in}}$-> $b \in \mathbb{R}^{d}$.
> > > > > >
> > > > > > Thank you so much for your careful reading! We have corrected it.

---

> ### Comment · Reviewer_ZrmK · 2022-11-20
> **Thanks for the replies!**
>
> The replies are very detailed and mostly addressed my concerns. I've raised my score to 8.

---

> > ### Author Response · Authors · 2022-11-22
> > **Thanks for your reply!**
> >
> > Thanks a lot! As mentioned in our last reply to your comments, we have also conducted one more ablation study in Section 4.4 (Page 8-9) on Text8 dataset to compare the proposed RSA module against another  modified XL (TL-XL) (i.e. $P_{\text{RSA}}$ is replaced by $P_{\text{TL}}$ with $P_{\text{TL}}$ being a Toeplitz learnable matrix).  The full experiment results are presented below, and for your easy reference, the newly added model is marked with "*".
> >
> > | #  layers |              | 8     | 10        | 12        | 14        | 16        |
> > | --------- | ------------ | ----- | --------- | --------- | --------- | --------- |
> > | XL        | #Params  (M) | /     | 34.14     | 39.42     | 47.79     | 54.61     |
> > |           | BPC          | /     | 1.196     | 1.184     | 1.171     | 1.170     |
> > | RSA-XL    | #Params  (M) | /     | 34.14     | 39.42     | 47.79     | 54.61     |
> > |           | BPC          | /     | **1.181** | **1.170** | **1.164** | **1.160** |
> > | FL-XL     | #Params  (M) | 35.72 | 44.65     | 53.58     | 62.51     | /         |
> > |           | BPC          | 1.214 | 1.196     | 1.189     | 1.182     | /         |
> > | *TL-XL    | #Params  (M) | /     | 34.18     | 41.01     | 47.85     | 54.68     |
> > |           | BPC          | /     | 1.193     | 1.188     | 1.183     | 1.178     |
> >
> > From the above table, it can be seen that (1) the newly added TL-XL also performs worse than the XL baseline of a similar model size, indicating parameter redundancy; (2) RSA-XL is still the most parameter-efficient design among the three models, namely it achieves the best performance when the number of parameters is comparable to all other models.
> >
> > This result will be further included into  Section 4.4 of the paper.

---

### Official Review · Reviewer_f1my · 2022-10-24

**Confidence:** 4
**Clarity, Quality, Novelty And Reproducibility:** The draft includes details for implem…
**Correctness:** 3
**Technical Novelty And Significance:** 3
**Empirical Novelty And Significance:** 3
**Recommendation:** 8

**Strength And Weaknesses:**

It is novel enough to combine the advantages of two famous models (Transformer, RNN). Also, the combining method looks applicable to a variety of scenarios. The experimental results are impressive, showing superior performance to previous Transformer.

I think the draft would become better if there is a more complete explanation and figures about the self-attention with recurrence (RSA) operation.


**Summary Of The Paper:**

Transformers have less inductive bias; while it is well generalized. RNNs have large inductive bias; while it is sample-efficient. Authors proposed the REM which combines the advantages of Transformer and RNNs which are famous sequential models. Experimental results have shown the effectiveness of the proposed model.


**Summary Of The Review:**

I think the novelty of this draft is enough for the publication and the experimental results are impressive. English is good enough as well. I recommend weak accept for the draft.

---

> ### Author Response · Authors · 2022-11-15
> **Thanks for your comments!**
>
> Thanks for your encouraging words and constructive comments. We sincerely appreciate your time in reading the paper, and our point-to-point responses to your comments are given below.
>
> > I think the draft would become better if there is a more complete explanation and figures about the self-attention with recurrence (RSA) operation.
>
> Thank you for this instructive comment. Following your suggestions, we have provided a graphical illustration of a single headed RSA module in Figure 1 (d) on Page 2, and a more detailed explanation about the operation of RSA has been given in the paragraph of "Operation of multihead RSA modules" on Page 5.
>
> In the meanwhile, we have also reorganized the whole Section 3 to better explain the proposed RSA. Specifically,
> For a single head RSA, we have devoted a paragraph right after equation (4) to detail the different types of REMs i.e. $\mathbf{P}$ in the paper.
>
> For your easy reference, we have listed the multihead RSA operation below:
>
> 	Procedure for the Multihead RSA
> 		- Choose masked or unmasked REMs according to the nature of the task.
> 		- Select the hyperparameters including the dilating factor $d$ and the numbers of the six types of REMs $(k_1,\dots,k_6)$.
> 		- For each head, apply equation (4) with a different REM.
> 		- Apply a linear layer to combine the output from all heads, and perform layer-normalization and dropout.

---

### Official Review · Reviewer_mvWh · 2022-10-25

**Confidence:** 5
**Correctness:** 3
**Technical Novelty And Significance:** 2
**Empirical Novelty And Significance:** 2
**Recommendation:** 6

**Clarity, Quality, Novelty And Reproducibility:**

Clarity: The paper is easy to read.

Quality: The paper tries to tackle an important problem.

Novelty: Even though the problem is not "new" per se, but the underlying idea is interesting.

Reproducibility: The paper should be easy to reproduce.

**Strength And Weaknesses:**

Strengths:

- The paper is easy to read, and generally well written.
- The paper evaluates the proposed method on various different tasks such as time-series forecasting, code and language modelling. The paper augments the proposed method to various different transformer variants and compares the performance with respect to the unmodified baseline.

Weakness:

- The problem of integrating recurrence and self-attention is an important research problem. There exists some existing ways on how to augment transformers with recurrence such as Temporal Latent Bottleneck [1] and Block-Recurrent Transformers [2]. The idea behind TLB is to "divide" the sequence into chunks, and within a chunk use self-attention and to access information across chunks the model needs to use recurrence. It would be useful to compare the proposed method to these variants to futher analyse the pros and cons.
- It may also be useful to study the proposed method by varying the capacity of the network to see how well the underlying idea scales.

[1] Temporal Latent Bottleneck: Synthesis of Fast and Slow Processing Mechanisms in Sequence Learning, https://arxiv.org/abs/2205.14794 (NeurIPS'22) \
[2] Block Recurrent Transformers, https://arxiv.org/abs/2203.07852 (NeurIPS'22)

**Summary Of The Paper:**

The paper tackles the problem of endowing Transformers with the ability to encode information about the past via recurrence. The proposed architecture can leverage the recurrent connections to improve the sample efficiency while maintaining expressivity due to the use of self-attention.

**Summary Of The Review:**

The paper proposes a way to incorporate recurrence and self-attention by modifying the positional encoding.

---

> ### Author Response · Authors · 2022-11-18
> **Thanks for your comment!**
>
> Thank you for your constructive comments and suggestions, and they are exceedingly helpful for us to improve our paper. We have carefully incorporated them in the revised paper. In the following, your comments are first stated and then followed by our point-by-point responses.
>
> > The problem of integrating recurrence and self-attention is an important research problem. There exists some existing ways on how to augment transformers with recurrence such as Temporal Latent Bottleneck [1] and Block-Recurrent Transformers [2]. The idea behind TLB is to "divide" the sequence into chunks, and within a chunk use self-attention and to access information across chunks the model needs to use recurrence. It would be useful to compare the proposed method to these variants to further analyze the pros and cons.
>
> Thanks very much for bringing to our attention the two important papers: Temporal Latent Bottleneck (TLB) [1] and Block-Recurrent Transformers (BRT) [2].
>
> On one hand, similar to Transformer-XL, TLB and BRT apply the recurrent operation and the self-attention operation to learn the coarse-grained and fine-grained temporal information of a sequence, respectively. Specifically, Transformer-XL partitions the long inputs into segments and then introduces recurrence along the sequence of segments. Meanwhile both TLB and BRT further divide each segment into smaller chunks (or blocks), and each chunk can be summarized into a few state vectors, which are updated by self-attention and/or cross-attention in a recurrent manner. In contrast, the proposed RSA uses both recurrent and self-attention operations to model the fine-grained token-level recurrence.
>
> On the other hand, both TLB and BRT take turns to apply the recurrent and self-attention operations, while the RSA manages to combine these two together into one single operation, by applying the findings in Proposition 2.
> As a result, the RSA module can be used to modify the self-attention of both TLB and BRT to further improve their performance.
>
> In fact, we have included BRT as a new baseline model for the natural language modeling tasks in Section 4.3. The self-attention operations within each chunk (or block) has been modified by using the proposed RSA module, and the resulting models are denoted by RSA-BRT, respectively.  However, since BRT's codes from [the official github repository](https://github.com/google-research/meliad) is written in Tensorflow. As a result, we have to spend a couple of days to write the PyTorch implementation of their model by ourselves. Due to limited resources, we have only obtained a few results up to now, and we present in the incomplete table as below for your easy reference.
>
> |                     | Enwik8     |        | WikiText-103 |        | Text8  |      |
> | ------------------- | ---------- | ------ | ------------ | ------ | ------ | ---- |
> |                     | Params     | BPC    | Params       | BPC    | Params | BPC  |
> | BRT                 | 55,555,788 | 1.0787 | 159,627,039  | 23.758 | *      | *    |
> | RSA-BRT             | 55,555,853 | 1.0746 | *            | *      | *      | *    |
> | Increase in #Params | 65         |        |              |        |        |      |
>
> *still running
>
> From the result for Enwik8, it can be seen that RSA-BRT achieves slightly better performance than its baseline model. Note that the difference is not as significant as the difference between RSA-XL against XL. Since a chunk is 4 times shorter than a segment, the additional benefit brought by the RSA may be smaller for the BRT baseline models. We are still running the remaining experiments for BRT and RSA-BRT, and will update you once the results are available. All results will be further included into Table 3 of the paper.

---

> > ### Author Response · Authors · 2022-11-18
> > **Additional reply to the first comment**
> >
> > Finally, following your suggestions, we have included TLB and BRT into the related work in Section 1.1 of the paper. More discussions on multiscale recurrence has been further provided in Appendix A, and a graphical illustration is also presented in Figure 5 for better clarification.
> >
> > In addition, we also made an attempt to implement the TLB model, and hence the RSA-TLB. While BRT performs recurrence on only one Transformer layer,TLB needs to iterate through all the layers before it updates the recurrent states. As a result, although TLB gains the benefit of incorporating high-level information from the latter layers, it requires much longer computation time. In the meanwhile, TLB's official codes from their supplementary material is not complete. We are trying our best to see if we would be able to provide you with some results on TLB and RSA-TLB afterwards.
> >
> > **Summary of paper revision related to this comment**:
> >
> > (1) We have modified the related works in Section 1.1 to include both TLB and BRT models.
> >
> > (2) We have provided more discussions on the multiscale recurrence in Appendix A with a graphical illustration in Figure 5.
> >
> > (3) We have included BRT as our baseline models in Section 4.3, and it has been further modified into RSA-BRT by using our RSA module. We will make updates once the results are available.

---

> > > ### Author Response · Authors · 2022-11-18
> > > **Reply to the second comment**
> > >
> > > > It may also be useful to study the proposed method by varying the capacity of the network to see how well the underlying idea scales.
> > >
> > > Thanks very much for this insightful comment. The proposed RSA module is very parsimonious, i.e. it requires only a few more parameters than the corresponding baseline model, and hence a model-scaling experiment can be very useful in illustrating the parameter efficiency of RSA.
> > >
> > > An ablation study has been conducted on Text8 dataset in Section 4.4, and we have varied the number of layers from 10 to 16 for Transformer-XL and the modified RSA-XL. The results are plotted in Figure 4 (left panel) of the paper, and we present the results in the table below for your easy reference.
> > >
> > > | #  layers           | 10         |           | 12         |           | 14         |           | 16         |           |
> > > | ------------------- | ---------- | --------- | ---------- | --------- | ---------- | --------- | ---------- | --------- |
> > > |                     | Params     | BPC       | Params     | BPC       | Params     | BPC       | Params     | BPC       |
> > > | XL                  | 34,139,675 | 1.196     | 40,964,635 | 1.184     | 47,789,595 | 1.171     | 54,614,555 | 1.170     |
> > > | RSA-XL              | 34,139,725 | **1.181** | 40,964,695 | **1.170** | 47,789,665 | **1.164** | 54,614,635 | **1.160** |
> > > | Increase in #Params | 50         |           | 60         |           | 70         |           | 80         |           |
> > >
> > > As the number of layers increases, the number of additional parameters in RSA-XL increases proportionally as expected. It can be seen that, with only less than 100 new parameters, RSA-XL can achieve significantly improvement over the baseline XL. More importantly, the advantage can be consistently observed for all model sizes.
> > >
> > > **Summary of paper revision related to this comment:**
> > >
> > > An ablation study has been conducted to study the parameter-efficiency of RSA, and has been added to Section 4.4 on Page 8-9 with results shown in Figure 4 (left panel).
> > >
> > >
> > >
> > > References
> > >
> > > [1] Didolkar, A. R., Gupta, K., Goyal, A., Lamb, A., Ke, N. R., and Bengio, Y. (2022). Temporal latent bottleneck: Synthesis of fast and slow processing mechanisms in sequence learning. In Advances in Neural Information Processing Systems.
> > >
> > > [2] Hutchins, D., Schlag, I., Wu, Y., Dyer, E., and Neyshabur, B. (2022). Block-recurrent transformers. In Advances in Neural Information Processing Systems.

---

> > > > ### Comment · Reviewer_mvWh · 2022-11-18
> > > > **Thank you**
> > > >
> > > > Dear. Authors,
> > > >
> > > > Thank you for taking time to run more experiments. The reviewer appreciates it.
> > > >
> > > > Summary of Changes:
> > > >
> > > > - Including the references in the paper: The reviewer thinks that authors should acknowledge that there's already existing work in integrating Transformers and recurrence like TLB and BRT in the introduction as compared to just mentioning in the related work.
> > > > - Running comparisons to Block Recurrent Transformers: The reviewer appreciates comparison to BRT, even though there's not much difference.
> > > > - Running scaling experiments: It's interesting to know that RSA-XL achieve  improvements over the baseline XL when we scale the number of layers. Similarly, running comparisons (RSA-BRT v/s BRT) and showing that RSA-BRT "scales" better as compared to BRT would be a useful experiment, and further improve the paper.

---

> > > > > ### Author Response · Authors · 2022-11-19
> > > > > **Thanks for your prompt reply**
> > > > >
> > > > > Thanks very much for your quick response with such detailed and constructive comments. We appreciate the effort very much and are ready to make further improvements to the paper following your advice.
> > > > >
> > > > >
> > > > >
> > > > > 1. Response to "including the references in the paper"
> > > > >
> > > > > Thanks for this detailed comment. We have made changes to the paper as you suggested. Specifically, we have added new paragraph to the Introduction to discuss the existing networks on integrating Transformers and recurrence (Page 2, Paragraph 3). For your easy reference, we quote the corresponding paragraphs below, and highlight our changes in bold letters.
> > > > >
> > > > >
> > > > >
> > > > > > **Recent efforts have been spent on integrating recurrence and self-attention systematically. Feedback Transformer [1] introduces the memory vectors to aggregate information across layers, and uses them to update the next token in a recursive manner. However, the computationally expensive sequential operation limits its attractiveness. Another line of research applies the recurrent operation only to aggregate the temporal information at a coarser scale, while the token-by-token dependence is learned by self-attention instead. Transformer-XL [2] partitions the long inputs into segments and introduces a segment-level recurrence. Meanwhile, Temporal Latent Bottleneck (TLB) [3] and Block-Recurrent Transformer (BRT) [4] further divide the segments into smaller chunks, and each chunk is summarized into a few state vectors. A recurrent relation is then formed on the sequence of state vectors.  These hierarchical designs are useful to reduce the computational burden, while they overlook recurrent dynamics at a finer scale.**
> > > > > >
> > > > > >
> > > > > >
> > > > > > In an attempt to simplify the numerical calculation of RNNs, we found surprisingly that an RNN layer with linear activation can be broken down into a series of simple RNNs with scalar hidden coefficients. Each simple RNN induces a distinct recurrent pattern, and their combination forms the recurrent dynamics of the RNN layer. Hence the calculation time can be greatly reduced by training these simple RNNs in parallel. On top of that, it can be equivalently rewritten into the positional encodings of a multihead self-attention (MHSA). This spontaneously inspires a solution, the multihead Self-Attention with Recurrence (RSA), to combine self-attention with RNN into one single operation while maintaining parallel computation. This solution enables our design to preserve the merits from both Transformer and recurrent models, while their respective shortcomings are avoided. **More importantly, it can be used to replace the self-attention of existing networks, such as Transformer XL, TLB and BRT, to further explore recurrent dynamics at the finer scale.** Our paper makes three main contributions below.
> > > > >
> > > > >
> > > > >
> > > > > 2. Response to "Running comparisons to Block Recurrent Transformers"
> > > > >
> > > > > Thanks for the comment, we will update you when the remaining results become available.
> > > > >
> > > > >
> > > > >
> > > > > 3. Response to "Running scaling experiments"
> > > > >
> > > > > Thanks for the constructive suggestion. It would indeed be interesting to compare the performance of RSA-BRT against its BRT baseline as the number of layers increases. We will conduct this experiment and update you when the results are obtained.
> > > > >
> > > > >
> > > > >
> > > > > References
> > > > >
> > > > > [1] Fan, A., Lavril, T., Grave, E., Joulin, A., and Sukhbaatar, S. (2021). Addressing some limitations of transformers with feedback memory. arXiv preprint arXiv:2002.09402.
> > > > >
> > > > > [2] Dai, Z., Yang, Z., Yang, Y., Carbonell, J. G., Le, Q., and Salakhutdinov, R. (2019). Transformer-xl: Attentive language models beyond a fixed-length context. In Proceedings of the 57th Annual Meeting of the Association for Computational Linguistics, pages 2978–2988.
> > > > >
> > > > > [3] Didolkar, A. R., Gupta, K., Goyal, A., Lamb, A., Ke, N. R., and Bengio, Y. (2022). Temporal latent bottleneck: Synthesis of fast and slow processing mechanisms in sequence learning. In Advances in Neural Information Processing Systems.
> > > > >
> > > > > [4] Hutchins, D., Schlag, I., Wu, Y., Dyer, E., and Neyshabur, B. (2022). Block-recurrent transformers. In Advances in Neural Information Processing Systems.

---

> > > > > > ### Author Response · Authors · 2022-11-24
> > > > > > **Thanks for your patience!**
> > > > > >
> > > > > > Thanks very much for your patience! To address your latest comments, we have continuously conducted experiments in the last few days, and has finally obtained the full results. Our point-to-point responses to your comments are given below.
> > > > > >
> > > > > > > - Running comparisons to Block Recurrent Transformers: The reviewer appreciates comparison to BRT, even though there's not much difference.
> > > > > >
> > > > > > We have completed running the experiments on RSA-BRT against its BRT baselines. Moreover, to address your concern, we tried varying the initial value for the gating parameter $\mu$ for the previous experiments on Enwik8 dataset. An improvement can be observed from the table below.
> > > > > >
> > > > > > |                     | Enwik8     |           | WikiText-103 |             | Text8      |        |
> > > > > > | ------------------- | ---------- | --------- | ------------ | ----------- | ---------- | ------ |
> > > > > > |                     | Params     | BPC       | Params       | BPC         | Params     | BPC    |
> > > > > > | BRT                 | 55,555,788 | 1.0746     | 159,627,039  | 23.758      | 55,464,987 | 1.1767  |
> > > > > > | RSA-BRT             | 55,555,853 | **1.0683** | 159,627,144  | **23.639** | 55,465,052 | **1.1758** |
> > > > > > | Increase in #Params | 65         |           | 105          |             | 65         |        |
> > > > > >
> > > > > >
> > > > > > It can be seen that RSA-BRT achieves better performance on all datasets.
> > > > > >
> > > > > >
> > > > > >
> > > > > > > - Running scaling experiments: It's interesting to know that RSA-XL achieve improvements over the baseline XL when we scale the number of layers. Similarly, running comparisons (RSA-BRT v/s BRT) and showing that RSA-BRT "scales" better as compared to BRT would be a useful experiment, and further improve the paper.
> > > > > >
> > > > > > Thanks again for this constructive suggestion. The scaling experiments have been conducted on the Enwik8 dataset, where each model is trained for 40 epochs (i.e. about 1e5 iterations). The results are presented in the table below. Note that in the previous table, each model is training for 2e5 iterations, and hence different BPCs for the 14-layer network can be observed.
> > > > > >
> > > > > > | #  layers           | 8          |           | 10         |           | 12         |           | 14         |           |
> > > > > > | ------------------- | ---------- | --------- | ---------- | --------- | ---------- | --------- | ---------- | --------- |
> > > > > > |                     | Params     | BPC       | Params     | BPC       | Params     | BPC       | Params     | BPC       |
> > > > > > | BRT                 | 35,080,908 | 1.127     | 41,905,868 | 1.106     | 48,730,828 | 1.098     | 55,555,788 | 1.079     |
> > > > > > | RSA-BRT             | 35,080,943 | **1.120** | 41,905,913 | **1.104** | 48,730,883 | **1.092** | 55,555,853 | **1.072** |
> > > > > > | Increase in #Params | 35         |           | 45         |           | 55         |           | 65         |           |
> > > > > >
> > > > > > Similarly, it can be seen that, with only less than 100 new parameters, RSA-BRT can achieve some improvement over the baseline BRT. More importantly, the advantage can be consistently observed for all model sizes.
> > > > > >
> > > > > > **Further paper revision**
> > > > > >
> > > > > > We are ready to add all the above results to Sections 4.3 and 4.4 of the paper.

---

> > > > > > > ### Author Response · Authors · 2022-11-30
> > > > > > > **Thanks again for your patience!**
> > > > > > >
> > > > > > > Thanks again for your constructive comments! After a few days of fine-tuning, we have updated the results for RSA-BRT against the BRT baseline in the table below.
> > > > > > >
> > > > > > > |                     | Enwik8     |            | WikiText-103 |            | Text8      |            |
> > > > > > > | ------------------- | ---------- | ---------- | ------------ | ---------- | ---------- | ---------- |
> > > > > > > |                     | Params     | BPC        | Params       | BPC        | Params     | BPC        |
> > > > > > > | BRT                 | 55,555,788 | 1.0746     | 159,627,039  | 23.758     | 55,464,987 | 1.1767     |
> > > > > > > | RSA-BRT             | 55,555,853 | **1.0683** | 159,627,144  | **23.639** | 55,465,052 | **1.1758** |
> > > > > > > | Increase in #Params | 65         |            | 105          |            | 65         |            |
> > > > > > >
> > > > > > > It can be observed that RSA-BRT can consistently achieve better performance than its baseline with merely100 more parameters. This table will be included into Section 4.4 of the paper, as well as all the helpful scaling experiments that you have suggested.
> > > > > > >
> > > > > > > We hope that we have by far addressed all your previous comments and concerns. And we are willing to respond to any further comments or suggestions.

---

> > > > > > > > ### Comment · Reviewer_mvWh · 2022-12-06
> > > > > > > > **Thank you**
> > > > > > > >
> > > > > > > > Dear. Authors,
> > > > > > > >
> > > > > > > > Thank you for making changes. I appreciate it.
> > > > > > > >
> > > > > > > > It seems that the proposed model seems to perform on-par with BRT (even if not better). I would encourage the authors to make further comparisons even on toy problems (like the one explored in TLB also) to better understand the differences between proposed method and how it compares to TLB and BRT.
> > > > > > > >
> > > > > > > > Thank you for running more experiments and for your time. I appreciate it.

---

> > > > > > > > > ### Author Response · Authors · 2022-12-07
> > > > > > > > > **Thanks so much for your reply and additional comments!**
> > > > > > > > >
> > > > > > > > > Thanks very much for your encouragement!
> > > > > > > > >
> > > > > > > > > Following your suggestions, we are conducting a traditional copy task (as in TLB) to make further comparisons between the proposed method and both the TLB and BRT models. Hopefully we can get back to you with some results before the end of the review period.

---

> > > > > > > > > > ### Author Response · Authors · 2022-12-11
> > > > > > > > > > **Thanks for your suggestion!**
> > > > > > > > > >
> > > > > > > > > > In the last few days, following your constructive advice, we have conducted a traditional copy task on four models: BRT, TLB, RSA-BRT and RSA-TLB, where RSA-BRT is obtained by simply changing the self-attention (SA) modules into RSA modules, and RSA-TLB is obtained similarly. Simply speaking, the proposed RSA module can leverage recurrent inductive bias of the REM part to achieve a better sample efficiency than its corresponding baseline Transformer, while its SA part can take care of the remaining signals. The following two experiments demonstrates the efficiency of our RSA modules from two aspects: (1) faster convergence during training, and (2) better sample efficiency.
> > > > > > > > > >
> > > > > > > > > > In the first experiment, we have used a slightly harder copy task from [1] (see also [2]). Specifically, the transformer has to copy a sequence of ten different symbols. The target sequence has a random length capped at 128, 256, 512 or 1024, respectively, in each training and test sample. We have followed the train settings and schemes in [2], and each model has been trained for 20 epochs. The results are presented in the table below.
> > > > > > > > > >
> > > > > > > > > > | Max sequence length | Model      | Accuracy(%) | Number of epochs needed to achieve 99%  training accuracy | Model      | Accuracy(%) | Number of epochs needed to achieve 99%  training accuracy |
> > > > > > > > > > | ------------------- | ---------- | ----------- | --------------------------------------------------------- | ---------- | ----------- | --------------------------------------------------------- |
> > > > > > > > > > | 128                 | BRT        | 99.75       | 7                                                         | TLB        | 99.50       | 13                                                        |
> > > > > > > > > > |                     | RSA-BRT    | **99.98**   | 2                                                         | RSA-TLB    | **99.82**   | 10                                                        |
> > > > > > > > > > |                     | Difference | 0.23        |                                                           | Difference | 0.31        |                                                           |
> > > > > > > > > > | 256                 | BRT        | 99.41       | 7                                                         | TLB        | 97.03       | 20                                                        |
> > > > > > > > > > |                     | RSA-BRT    | **99.99**   | 2                                                         | RSA-TLB    | **99.74**   | 11                                                        |
> > > > > > > > > > |                     | Difference | 0.58        |                                                           | Difference | 2.71        |                                                           |
> > > > > > > > > > | 512                 | BRT        | 99.25       | 9                                                         | TLB        | 99.50       | 14                                                        |
> > > > > > > > > > |                     | RSA-BRT    | **99.99**   | 2                                                         | RSA-TLB    | **99.91**   | 4                                                         |
> > > > > > > > > > |                     | Difference | 0.74        |                                                           | Difference | 0.41        |                                                           |
> > > > > > > > > > | 1024                | BRT        | 99.43       | 6                                                         | TLB        | 99.69       | 14                                                        |
> > > > > > > > > > |                     | RSA-BRT    | **100.00**  | 2                                                         | RSA-TLB    | **99.97**   | 4                                                         |
> > > > > > > > > > |                     | Difference | 0.56        |                                                           | Difference | 0.28        |                                                           |
> > > > > > > > > >
> > > > > > > > > > It can be seen that all models achieve very high accuracy for this synthetic task. Even so, RSA models consistently outperform their respective baselines (BRT vs RSA-BRT, TLB vs RSA-TLB) under each setting. In addition, by plotting the loss trajectory during training, it can be observed that RSA models converge much faster than their baselines.

---

> > > > > > > > > > > ### Author Response · Authors · 2022-12-11
> > > > > > > > > > > **Second experiment on copy task**
> > > > > > > > > > >
> > > > > > > > > > > In the second experiment, we have further reduced the samples size of training samples from 12800 (in the first experiment) to 6400. The corresponding results are presented below.
> > > > > > > > > > >
> > > > > > > > > > > | Number of  training samples | Max sequence length | Model      | Accuracy(%) | Model      | Accuracy(%) |
> > > > > > > > > > > | --------------------------- | ------------------- | ---------- | ----------- | ---------- | ----------- |
> > > > > > > > > > > | 6400                        | 256                 | BRT        | 96.63       | TLB        | 87.06       |
> > > > > > > > > > > |                             |                     | RSA-BRT    | **99.96**   | RSA-TLB    | **98.96**   |
> > > > > > > > > > > |                             |                     | Difference | 3.34        | Difference | 11.90       |
> > > > > > > > > > > |                             | 512                 | BRT        | 98.08       | TLB        | 96.64       |
> > > > > > > > > > > |                             |                     | RSA-BRT    | **99.99**   | RSA-TLB    | **99.75**   |
> > > > > > > > > > > |                             |                     | Difference | 1.91        | Difference | 3.11        |
> > > > > > > > > > > |                             | 1024                | BRT        | 98.37       | TLB        | 99.17       |
> > > > > > > > > > > |                             |                     | RSA-BRT    | **99.99**   | RSA-TLB    | **99.79**   |
> > > > > > > > > > > |                             |                     | Difference | 1.63        | Difference | 0.62        |
> > > > > > > > > > >
> > > > > > > > > > > It can be observed that when the sample size decreases, the prediction accuracy for both baseline models, BRT and TLB, drops, while the two RSA models can still achieve near perfect accuracy. This gives further supports to the sample efficiency of RSA designs.
> > > > > > > > > > >
> > > > > > > > > > >
> > > > > > > > > > >
> > > > > > > > > > > [1] Nguyen, Tan, et al. "Fmmformer: Efficient and flexible transformer via decomposed near-field and far-field attention." *Advances in neural information processing systems* 34 (2021): 29449-29463.
> > > > > > > > > > >
> > > > > > > > > > > [2] Katharopoulos, Angelos, et al. "Transformers are rnns: Fast autoregressive transformers with linear attention." *International Conference on Machine Learning*. PMLR, 2020.

---

> > > > > ### Author Response · Authors · 2022-12-04
> > > > > **Another update on experiment and looking forward to your comments!**
> > > > >
> > > > > In the last three days, we have made an effort to fine-tune the results of both BRT and RSA-BRT on the Text8 dataset, and have achieved better performance for both models, as presented in the table below.
> > > > > |                     | Text8      |            |
> > > > > | :------------------ | :--------- | :--------- |
> > > > > |                     | Params     | BPC        |
> > > > > | BRT                 | 55,464,987 | 1.1652     |
> > > > > | RSA-BRT             | 55,465,052 | **1.1625** |
> > > > > | Increase in #Params | 65         |            |
> > > > >
> > > > > While significant improvement can be observed for both models, it can be seen that RSA-BRT still outperforms BRT.
> > > > >
> > > > > ## A summary of all changes in phase II of Discussion
> > > > >
> > > > > Up till now, we have addressed all of your most recent comments. We will summarize them here point-to-point for your easy reference:
> > > > >
> > > > > > - Including the references in the paper: The reviewer thinks that authors should acknowledge that there's already existing work in integrating Transformers and recurrence like TLB and BRT in the introduction as compared to just mentioning in the related work.
> > > > >
> > > > > **Summary of change.** We have added new paragraph to the Introduction to discuss the existing networks on integrating Transformers and recurrence (Page 2, Paragraph 3).
> > > > >
> > > > > > - Running comparisons to Block Recurrent Transformers: The reviewer appreciates comparison to BRT, even though there's not much difference.
> > > > >
> > > > > **New experiment results.** We have completed the experiments of RSA-BRT vs BRT. With some fine-tuning, it can be observed from that RSA-BRT can achieve much better results than BRT. Considering that RSA-BRT only requires approximately 100 additional parameters than RSA-baseline, this improvement is quite promising. The newest and complete table is presented below, and we are ready to include it into the paper.
> > > > >
> > > > > |                     | Enwik8     |            | WikiText-103 |            | Text8      |            |
> > > > > | :------------------ | :--------- | :--------- | :----------- | :--------- | :--------- | :--------- |
> > > > > |                     | Params     | BPC        | Params       | BPC        | Params     | BPC        |
> > > > > | BRT                 | 55,555,788 | 1.0746     | 159,627,039  | 23.758     | 55,464,987 | 1.1652     |
> > > > > | RSA-BRT             | 55,555,853 | **1.0683** | 159,627,144  | **23.639** | 55,465,052 | **1.1625** |
> > > > > | Increase in #Params | 65         |            | 105          |            | 65         |            |
> > > > >
> > > > > Thanks again for your valuable suggestion on conducting this experiment. It is indeed very helpful for solidifying the effectiveness and general applicability of the proposed RSA module.
> > > > >
> > > > > > - Running scaling experiments: It's interesting to know that RSA-XL achieve improvements over the baseline XL when we scale the number of layers. Similarly, running comparisons (RSA-BRT v/s BRT) and showing that RSA-BRT "scales" better as compared to BRT would be a useful experiment, and further improve the paper.
> > > > >
> > > > > **New experiment results.** We have performed the scaling experiments. And similarly, it is observed that RSA-BRT indeed scales better as compared to BRT. More details for the experiments are omitted here but contained in the post dated 30 Nov 2022, if you are interested. We attach the results below for your easy reference.
> > > > >
> > > > > | #  layers           | 8          |           | 10         |           | 12         |           | 14         |           |
> > > > > | ------------------- | ---------- | --------- | ---------- | --------- | ---------- | --------- | ---------- | --------- |
> > > > > |                     | Params     | BPC       | Params     | BPC       | Params     | BPC       | Params     | BPC       |
> > > > > | BRT                 | 35,080,908 | 1.127     | 41,905,868 | 1.106     | 48,730,828 | 1.098     | 55,555,788 | 1.079     |
> > > > > | RSA-BRT             | 35,080,943 | **1.120** | 41,905,913 | **1.104** | 48,730,883 | **1.092** | 55,555,853 | **1.072** |
> > > > > | Increase in #Params | 35         |           | 45         |           | 55         |           | 65         |           |
> > > > >
> > > > > The advantage can be consistently observed for all model sizes. We are ready to include this into Section 4.4.
> > > > >
> > > > > **We are really looking forward to hearing your views on the new experiments and changes!**

---

### Author Response · Authors · 2022-11-30
**For all reviewers: further paper revision**

We will make the following revisions to the paper:

1. Block-Recurrent Transformer (BRT) [1] has been adopted as another baseline model for the NLP experiment in Section 4.3, and its results are presented as follows.

|                             | BRT        | RSA-BRT    |
| --------------------------- | ---------- | ---------- |
| Enwik8                      | 1.0746     | **1.0683** |
| Text8                       | 1.1652 | **1.1625** |
| WikiText-103                | 23.758     | **23.639** |
| # Averaged Params added (%) |            | 8.68E-05   |

It can be seen that RSA-BRT exceeds the baseline BRT's performance on all datasets.

**The results of this table will be used to fill in the blanks in Table 3 (b) of the paper.**



2. Two additional experiments for Section 4.4 have been conducted during the second discussion phase, which are detailed in the responses to Reviewers mvWh and Zrmk.

   (1) A scaling experiment is conducted for RSA-BRT v/s BRT on Enwik8 dataset. The results are shown as follows.

   | #  layers           | 8          |           | 10         |           | 12         |           | 14         |           |
   | ------------------- | ---------- | --------- | ---------- | --------- | ---------- | --------- | ---------- | --------- |
   |                     | Params     | BPC       | Params     | BPC       | Params     | BPC       | Params     | BPC       |
   | BRT                 | 35,080,908 | 1.127     | 41,905,868 | 1.106     | 48,730,828 | 1.098     | 55,555,788 | 1.079     |
   | RSA-BRT             | 35,080,943 | **1.120** | 41,905,913 | **1.104** | 48,730,883 | **1.092** | 55,555,853 | **1.072** |
   | Increase in #Params | 35         |           | 45         |           | 55         |           | 65         |           |

   It can be seen that, with only less than 100 new parameters, RSA-BRT can achieve some improvement over the baseline BRT. More importantly, the advantage can be consistently observed for all model sizes.



   (2) Another scaling experiment is conducted for RSA-XL against TL-XL on Text8 dataset, where REM is replaced by a learnable Toeplitz matrix in the latter model. The results are shown as follows.

   | #  layers           | 8          |           | 10         |           | 12         |           | 14         |           |
   | ------------------- | ---------- | --------- | ---------- | --------- | ---------- | --------- | ---------- | --------- |
   |                     | Params     | BPC       | Params     | BPC       | Params     | BPC       | Params     | BPC       |
   | TL-XL               | 34,180,645 | 1.193     | 41,013,799 | 1.188     | 47,846,953 | 1.183     | 54,680,107 | 1.178     |
   | RSA-XL              | 34,139,725 | **1.181** | 40,964,695 | **1.170** | 47,789,665 | **1.164** | 54,614,635 | **1.160** |
   | Decrease in #Params | 40,920     |           | 49,104     |           | 57,288     |           | 65,472     |           |

   From the above table, it can be seen that the newly added TL-XL also performs worse than the RSA-XL of a similar model size, indicating parameter redundancy. In other words, RSA-XL enjoys a much better parameter-efficiency.

**These two experiments will be further included into Section 4.4 of the paper.**



Reference

[1] Hutchins, D., Schlag, I., Wu, Y., Dyer, E., and Neyshabur, B. (2022). Block-recurrent transformers. In Advances in Neural Information Processing Systems.

---

### Public Comment · ~Lokesh_Veeramacheneni1 · 2023-06-30
**a small question**

I would like to request further clarification regarding your paper after carefully reading it. Firstly, I would like to express my sincere appreciation for the captivating nature of your work and the clarity with which it is presented. Congratulations for the acceptance of your paper into the top 5% category.

In Section 4.3, I noticed the utilization of Transformer-XL with 14 layers, resulting in a notable achievement of 1.074 on the Enwik8 dataset. However, upon referencing the Transformer-XL paper, it became apparent that they reported lower bpc values, specifically 1.06 with 12 layers, 1.03 bpc with 18 layers, and an impressive 0.99 bpc with 24 layers.

To enhance my understanding, I kindly request your insights regarding the decision to opt for 14 layers instead and the possible reasons behind the relatively higher bpc despite employing deeper layers. Additionally, I would greatly appreciate any additional details or insights you can provide to address these inquiries.

Thank you in advance for your time and consideration. Your input will greatly contribute to my comprehension of your valuable research. Once again, congratulations on the successful publication of your paper.

---

> ### Author Response · Authors · 2023-07-09
> **Thanks for the question**
>
> Hi Lokesh, thanks for the question!
>
> The observed difference between the reported bits per character (bpc) for Enwik8 in Section 4.3 of our paper and the original Transformer-XL paper can be attributed to our decision to utilize Nvidia's implemented Transformer-XL (https://catalog.ngc.nvidia.com/orgs/nvidia/resources/transformerxl_for_pytorch) rather than the official repository. We chose the Nvidia version due to its enhanced user-friendliness and comprehensive multi-card support.
>
> However, it is important to note that the reproduction by Nvidia resulted in slightly worse bpc for Enwik8 compared to the figures reported in the original paper. Specifically, the bpc for Enwik8 with a 12-layer Transformer-XL exceeded the previously reported value of 1.06. This discrepancy could be attributed to variations in the implementation and environment between Nvidia's version and the official repository.
>
> Furthermore, from an intuitive perspective, when a model is overparameterized, the proposed RSA may exhibit better generalization ability, as illustrated in Figure 1. In order to emphasize the benefits of the proposed RSA, we employed a slightly larger model. Unfortunately, due to limited resources, we were unable to conduct further experiments using a 24-layer XL model.
>
> While acknowledging these limitations, we believe that the use of Nvidia's implementation, combined with our modifications, provides valuable insights and supports our argument. The comparison between the modified models, despite the slight deviations, offers meaningful observations regarding the potential advantages of the proposed RSA.

---

### Decision · Program_Chairs · 2023-01-20

**Decision:**

Accept: notable-top-5%

**Justification For Why Not Higher Score:**

N/A

**Justification For Why Not Lower Score:**

Reviewers unanimously recommended the acceptance of the paper with an average grade of 7.33 and strong average confidence of 4. The proposed solution has the potential to be impactful as it integrates recurrence in self-attention in a very principled way. As stated in Section V.1 of the meta-review, the presented solution has great potential to be of benefit to the whole community. The application, extension, and investigation of other models based on the presented solution can also be of great benefit to the representation learning community in general.




**Metareview: Summary, Strengths And Weaknesses:**


I Summary:

I.1 Investigated Problem:

Transformers models have the capacity to process large-scale sequential data and tend to overfit on small sequences. At the same time, RNNs inherently possess inductive bias that prevents overfitting and their training can be longer due to their inherent recurrence which hinders the leverage of modern-day parallelism of processing units like GPUs and TPUs.

- I.2 Proposed Solution:
The Recurrence Encoding Matrix (REM) is proposed to endow positional encodings of a multi-head self-attention with recurrent dynamics leading to a new module named Self-Attention with Recurrence (RSA). The proposed module can leverage the recurrent inductive bias of REMs to achieve a better sample efficiency than its corresponding baseline Transformer, while self-attention is used to model the remaining non-recurrent signals. The relative proportions between the RNN and the transformer are controlled by a data-driven gated mechanism supported by significantly improved performance.

- I.3 Validity Proof of the Proposed Solution:
    - Extensive experiment setting showcase the effectiveness of RSA modules demonstrated by four sequential learning tasks namely:
        - Time series forecasting;
        - Regular language learning;
        - Code language modelling;
        - Natural language modeling.
    - Transformers are augmented to various variants and compared with unmodified benchmarks and Block-Recurrent Transformers (BRT) which integrate recurrence and self-attention mechanisms. The conducted evaluation demonstrates the superiority of the presented method.

II Strengths:

II.1 From a structural (organization) point of view:
- The set is well-structured;
- The method is Cleary presented with descriptive figures.

II.2 From an analytical (development) point of view:
- The motivation is Clearly presented;
- Experimental setting confirms the benefit of the proposed as the superiority of the solution is illustrated by empirical evidence;
- Theoretical evidence is provided for the design of the solution ;
- The discussion related to the comparison conducted with several features of existing methods is appreciated.

II.3 From a perspective of soundness (unity, and coherence) and completeness (correctness):
- The strength points mentioned above are sufficient evidence of the soundness and completeness of the paper.
- An additional point reinforcing the strengths mentioned above is the active interaction of the authors during the rebuttal period and their openness to concerns and questions raised by the reviewers. The openness followed by the active interactions and persistence in answering questions and addressing concerns related to the submission demonstrate the author's intellectual honesty and good faith in conveying the details of the proposed solution to the reader as well as its benefits. Such a rebuttal is a great example of how rebuttals can be conducted to reinforce the value of the presented solution in the paper.

III Addressing what can be thought of as weaknesses:

- There are almost no weaknesses to point out. All of the concerns raised by the reviewers were addressed. Something to point out to the authors is the density aspect of the text. The theory behind the design of the method is smoothly introduced and an attentive reader could appreciate this. Yet, the appearance of the presented equations, lemmas, and propositions is dense.
- Most of the points that could be thought of as weaknesses have been and the reviewers agree Unanimously on the acceptance of the submission.

IV. Potential of the paper:

- IV.1 From a Potential perspective (Potential of the paper to the community): The proposed solution has a great potential to be of benefit to the whole community. The application, extension, and investigation of other models based on the presented solution can also be of great benefit to the representation learning community in general.




**Note From Pc:**

if the above contains the word "oral" or "spotlight" please see: "oral" presentation means -> notable-top-5% and "spotlight" means -> notable-top-25%. As stated in our emails, we are disassociating presentation type from AC recommendations

**Summary Of Ac-Reviewer Meeting:**

N/A